# High potency of sequential therapy with only β-lactam antibiotics

**Aditi Batra[1,2†], Roderich Roemhild[1,2,3†], Emilie Rousseau[4], Sören Franzenburg[5], Stefan Niemann[4], Hinrich Schulenburg[1,2]***

[1]Department of Evolutionary Ecology and Genetics, University of Kiel, Kiel, Germany; [2]Max Planck Institute for Evolutionary Biology, Ploen, Germany; [3]Institute of Science and Technology, Klosterneuburg, Austria; [4]Borstel Research Centre, National Reference Center for Mycobacteria, Borstel, Germany; [5]Competence Centre for Genomic Analysis Kiel, University of Kiel, Kiel, Germany

**Abstract** Evolutionary adaptation is a major source of antibiotic resistance in bacterial pathogens. Evolution-informed therapy aims to constrain resistance by accounting for bacterial evolvability. Sequential treatments with antibiotics that target different bacterial processes were previously shown to limit adaptation through genetic resistance trade-offs and negative hysteresis. Treatment with homogeneous sets of antibiotics is generally viewed to be disadvantageous as it should rapidly lead to cross-resistance. We here challenged this assumption by determining the evolutionary response of *Pseudomonas aeruginosa* to experimental sequential treatments involving both heterogenous and homogeneous antibiotic sets. To our surprise, we found that fast switching between only β-lactam antibiotics resulted in increased extinction of bacterial populations. We demonstrate that extinction is favored by low rates of spontaneous resistance emergence and low levels of spontaneous cross-resistance among the antibiotics in sequence. The uncovered principles may help to guide the optimized use of available antibiotics in highly potent, evolution-informed treatment designs.

*For correspondence:
hschulenburg@zoologie.uni-kiel.de

†These authors contributed equally to this work

Competing interests: The authors declare that no competing interests exist.

## Introduction

The efficacy of antibiotics for the treatment of infections is diminishing rapidly as bacteria evolve new mechanisms to resist antibiotics (*Laxminarayan et al., 2013*). Resistance evolution is frequently observed during antibiotic therapy and can happen within days (*Bloemberg et al., 2015*; *Hjort et al., 2020*; *Tueffers et al., 2019*). A failure to account for such rapid bacterial adaptation is likely a common reason for treatment failure (*Woods and Read, 2015*; *Zhou et al., 2020*). For this reason, the field of evolutionary medicine specifically accounts for bacterial evolvability and seeks treatment solutions that are hard to overcome by genetic adaptation (*Andersson et al., 2020*; *Merker et al., 2020*). While an evolution-proof antibiotic remains to be found, the mechanisms that restrict evolutionary escape are starting to be revealed (*Bell and MacLean, 2018*). Such evolutionary insight may guide the design of effective and sustainable antibiotic therapy.

An effective way of reducing the amount of evolutionary solutions is to administer several antibiotics either simultaneously (i.e., combination therapy) or sequentially (i.e., sequential therapy). Tailored combination treatments make use of physiological and evolutionary constraints (*Baym et al., 2016*). The emergence of resistance is delayed by combinations, when evolutionary escape requires multiple mutations and when drug interactions eliminate the intermediate genetic steps of single-drug resistance (*Chait et al., 2007*), antibiotic tolerance (*Levin-Reisman et al., 2017*), and heterore-sistance (*Band et al., 2019*). However, when genetic resistance to the combination is easily accessible, for example, through gene amplification of efflux pumps, then combination therapy can accelerate resistance emergence (*Pena-Miller et al., 2013*). This undesired selective effect is

**eLife digest** Overuse of antibiotic drugs is leading to the appearance of antibiotic-resistant bacteria; this is, bacteria with mutations that allow them to survive treatment with specific antibiotics. This has made some bacterial infections difficult or impossible to treat. Learning more about how bacteria evolve resistance to antibiotics could help scientists find ways to prevent it and develop more effective treatments.

Changing antibiotics frequently may be one way to prevent bacteria from evolving resistance. That way if a bacterium acquires mutations that allow it to escape one antibiotic, another antibiotic will kill it, stopping it from dividing and preventing the appearance of descendants with resistance to several antibiotics. In order to use this approach, testing is needed to find the best sequences of antibiotics to apply and the optimal timings of treatment.

To find out more, Batra, Roemhild et al. grew bacteria in the laboratory and exposed them to different sequences of antibiotics, switching antibiotics at different time intervals. This showed that sequential treatments with different antibiotics can limit bacterial evolution, especially when antibiotics are switched quickly. Unexpectedly, one of the most effective sequences used very similar antibiotics. This was surprising because using similar antibiotics should lead to the evolution of cross-resistance, which is when a drug causes changes that make the bacterium less sensitive to other treatments. However, in the tested case, cross-resistance did not evolve when antibiotics were switched quickly, thereby ensuring efficiency of treatment.

Batra et al. show that alternating sequences of antibiotics may be an effective strategy to prevent drug resistance. Because the experiments were done in a laboratory setting it will be important to verify the results in studies in animals and humans before the approach can be used in medical or veterinary settings. If the results are confirmed, it could reduce the need to develop new antibiotics, which is expensive and time consuming.

potentially avoided by sequential drug application. Evolutionary escape from sequential treatments is constrained by negative hysteresis responses induced by specific antibiotics (*Roemhild et al., 2018*) and/or the emergence of genetic collateral sensitivity trade-offs (*Barbosa et al., 2019*; *Yoshida et al., 2017*). Negative hysteresis occurs when exposure to an antibiotic induces changes to bacterial physiology that transiently increase the killing efficacy of other antibiotics (*Roemhild et al., 2018*). Collateral sensitivity is a genetic side effect of evolved resistance that too increases the efficacy of other antibiotics (*Szybalski and Bryson, 1952*). Collateral sensitivity is prevalent among pathogens and occurs especially between antibiotics with distinct mechanism of action (i.e., heterogeneous sets of antibiotics), while cross-resistance often emerges towards antibiotics with similar mode of action (i.e., homogeneous sets of antibiotics) (*Barbosa et al., 2017*; *Imamovic and Sommer, 2013*; *Lázár et al., 2013*; *Maltas and Wood, 2019*). Thus, conventionally, multidrug treatments would avoid antibiotics from similar classes, with the rationale of limiting the overlap in the respective sets of resistance mutations, and thus the ensuing cross-resistance.

The particular efficacy of sequential therapy has been confirmed with the help of evolution experiments under controlled laboratory conditions. Different types of sequential treatments have been tested. Some regimens involved a single switch between antibiotics, while others included multiple switches at short time intervals. One of the main findings was that the efficacy of sequential treatments depended both on the included antibiotics and the particular treatment sequence (*Fuentes-Hernandez et al., 2015*; *Maltas and Wood, 2019*; *Roemhild et al., 2015*). While fast sequential treatments did not exclude the eventual emergence of multidrug resistance, many significantly delayed bacterial adaptation compared to monotherapy (*Kim et al., 2014*; *Roemhild et al., 2015*; *Yoshida et al., 2017*). A single antibiotic switch can also delay adaptation, dependent on the drug order, and it can additionally reverse previous resistance and resensitize bacterial populations to specific antibiotics (*Barbosa et al., 2019*; *Hernando-Amado et al., 2020*; *Imamovic and Sommer, 2013*; *Yen and Papin, 2017*). Moreover, our group previously demonstrated that fast sequential treatments with a heterogeneous set of three antibiotics – the fluoroquinolone ciprofloxacin (CIP), the β-lactam carbenicillin (CAR), and the aminoglycoside gentamicin (GEN) – delayed the emergence of multidrug resistance in the pathogen *Pseudomonas aeruginosa* (*Roemhild et al., 2018*). The

observed inhibition of evolutionary escape was manifested by the occurrence of population extinction, although antibiotic concentrations were below the minimal inhibitory concentration (MIC). We further found that negative hysteresis at antibiotic switches reduced adaptation rates because it selected for distinct genetic changes. Several populations adapted to fast sequential treatment by independent mutations in the histidine kinase *cpxS* that only mildly increased resistance, thereby explaining the low rate of adaptation to the used antibiotics. Instead, the *cpxS* mutations suppressed negative hysteresis, demonstrating that adaptation was specific to the selective constraint imposed by the drug switches. Based on these findings, we assumed that the acting selective dynamics were ultimately a consequence of antibiotic heterogeneity. However, is this so? Do selective dynamics differ for a homogenous set of drugs?

The primary aim of our current study was to assess the efficacy of sequential treatments with either heterogeneous or homogeneous sets of three antibiotics. We focused on *P. aeruginosa* strain PA14 as a tractable pathogen model system, for which comprehensive experimental reference data is available on resistance evolution (e.g., *Barbosa et al., 2019*; *Barbosa et al., 2018*; *Hernando-Amado et al., 2020*; *Roemhild et al., 2018*; *Sanz-García et al., 2018*; *Yen and Papin, 2017*). We performed similar evolution experiments as before, with three new sets of bactericidal antibiotics, two of which included only β-lactams, and one the three previously considered modes of action (*Figure 1—figure supplement 1A*). The new heterogeneous drug set CIP, streptomycin (STR), and doripenem (DOR) involved drug synergy and was expected to contribute to collateral sensitivity (*Barbosa et al., 2018*; *Barbosa et al., 2017*). The drug sets comprising three β-lactams, however, had all properties that would typically be avoided for the design of multidrug treatments. The three β-lactams CAR, cefsulodin (CEF), and DOR have the same core structure and individually inhibit the DD-transpeptidase activity in cell-wall synthesis (*Walsh, 2003*). The collateral effects landscape between CAR-CEF-DOR was expected to be dominated by cross-resistance (*Barbosa et al., 2017*) and the three antibiotics showed neither synergy nor antagonism (*Barbosa et al., 2018*). Resistance to these antibiotics may potentially be achieved through single mutations. The situation is replicated by the set of ticarcillin (TIC), azlocillin (AZL), and ceftazidime (CEZ). In contrast to expectations, the triple β-lactam sequences showed high treatment potency. Therefore, the secondary aim of our study was to assess which characteristics constrained the ability of the bacteria to adapt to the β-lactam sequential treatments. We focused on one triple β-lactam set (CAR-CEF-DOR) and specifically tested the influence of antibiotic switching rate, switching regularity, negative hysteresis, the potential for spontaneous resistance evolution, and resulting cross-resistances on treatment efficacy.

## Results

### Triple β-lactam sequential treatments favor extinction of bacterial populations

We challenged a total of 756 replicate *P. aeruginosa* populations with sequential treatments across three fully independent evolution experiments, each focused on a different set of three antibiotics (*Figure 1*, *Figure 1—figure supplement 1*, *Supplementary file 1A*, Materials and methods). The antibiotic concentrations were calibrated to an inhibitory concentration of 75% (IC75), allowing bacteria to adapt to the imposed selection pressure. We used a serial dilution protocol for experimental evolution, with 2% culture transfer after 12 hr (one transfer) across a total of 96 transfers, equivalent to approximately 500 bacterial generations. Following the previous setup (*Roemhild et al., 2018*), we recorded the evolutionary dynamics in response to 16 different treatments, belonging to four main treatment types: monotherapy, fast-regular, slow-regular, and random sequential therapy (*Figure 1*).

Extinction of experimental populations differed considerably between the antibiotic sets. The two β-lactam sets produced a surprisingly high degree of extinction (CAR-CEF-DOR and TIC-AZL-CEZ; extinct fraction 27.2 and 13.3%, respectively, *Figure 1C*). The observed extinction frequency was comparable to that observed in the previous experiment with CAR-CIP-GEN (extinct fraction 15%, *Figure 1C*). CIP-DOR-STR caused no extinction, indicating that extinction was not explained by applying heterogeneous sets of antibiotics. Within the β-lactam sequential treatments, we observed that treatments that switched between antibiotics fast (every transfer) produced much higher extinction levels than those that switched slowly (every four transfers) or not at all (*Figure 1D*). Most of the

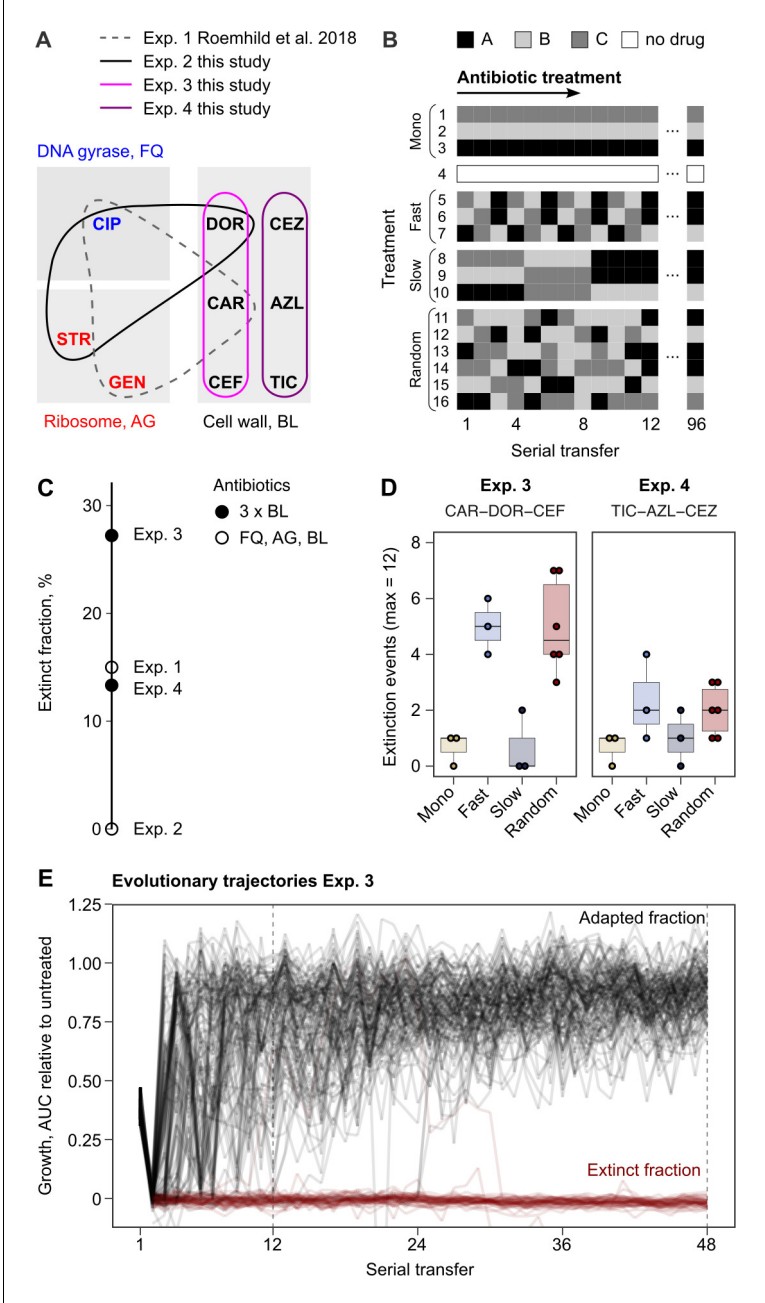

**Figure 1.** Probability of evolutionary rescue depends on drug triplets and treatment type. (**A**) The evaluated antibiotic combinations comprise different types of antibiotic targets. Fluoroquinolone antibiotics (FQ) target DNA gyrase, aminoglycosides (AG) inhibit translation, and β-lactams (BL) inhibit cell-wall synthesis. (**B**) The evaluated treatment protocols test the effects of switching rate and temporal regularity. (**C**) A fraction of lineages is eradicated by the sublethal dosage sequential treatments. Lineage extinction is high for combinations of cell-wall targeting β-lactams. (**D**) Variation in extinction for the β-lactam combinations by treatment type (n = 3–6 protocols per treatment type). (**E**) The distribution of evolutionary trajectories for Exp. 3 with CAR-DOR-CEF shows that the majority of extinction events occur within the first 12 serial transfers (n = 180 lineages). Growth of evolving lineages is quantified relative to untreated reference populations using the relative area under the growth curve (AUC). AZL: azlocillin; CAR: carbenicillin; CEF: cefsulodin; CEZ: ceftazidime; CIP: ciprofloxacin; DOR: doripenem; GEN: gentamicin; STR: streptomycin; TIC: ticarcillin. The following supplementary material is available for Figure 1: *Figure 1—figure supplement 1*, *Figure 1—source data 1*, *Figure 1—figure supplement 1—source data 1*, *Supplementary file 1A*.

The online version of this article includes the following source data and figure supplement(s) for figure 1:

*Figure 1 continued on next page*

*Figure 1 continued*

**Source data 1.** Source data for the panels of *Figure 1*.
**Figure supplement 1.** Antibiotic dose-response curves for PA14 (mean ± s.d.; n = 6 biological replicates).
**Figure supplement 1—source data 1.** Source data for *Figure 1—figure supplement 1*.

extinction events happened early in the experiment (*Figure 1E*), indicating that the initial treatment steps are critical for adaptation of populations. We conclude that fast sequential β-lactam treatments showed a surprising ability to restrict bacterial adaptation. As this result was unexpected, we decided to research the mechanisms that constrain resistance emergence in β-lactam sequences. Given that the experiment involving CAR-CEF-DOR produced the highest fraction of extinct populations, we decided to focus further analyses on this set.

## Resistance to doripenem was constrained in both monotherapy and switching treatments in the CAR-CEF-DOR triple β-lactam experiment

The CAR-CEF-DOR triple β-lactam experiment was characterized in detail for changes in growth, evolved resistance, and whole-genome sequences in order to assess the selection dynamics involved. We calculated the relative growth yield (see Materials and methods) at the end of each transfer and found growth dynamics to be divided into three phases: an early phase of rapid adaptation (transfers 1–12), followed by a phase of gradual growth yield convergence (transfers 13–48), and a final plateau phase (transfers 49–96) (*Figure 2A*; the growth phases are separated by vertical dotted lines). We compared the main treatment types using general linear models (GLM) for each phase separately (this fulfills the model assumption of response linearity). The early phase dynamics were characterized by significantly decelerated adaptation dynamics of the fast-regular group compared with monotherapy and slow-regular (GLM, post hoc test, p<0.037, *Supplementary file 1B*), but not random treatments. The slow-regular treatment did not differ significantly from monotherapy or random treatments (GLM, post hoc test, p=0.469, *Supplementary file 1B*). In the subsequent phase, growth yields of the groups converged to a plateau of roughly 90% relative yield, indicating similar final levels of adaptation (the growth yields of main treatment groups showed no statistical differences in phases 2 and 3, *Supplementary file 1B*). Alternating between the β-lactams fast and in a regular order therefore significantly constrained the growth of the bacterial populations. Intriguingly, in these fast sequential treatments, bacterial growth in the transfers with DOR was lower than in the transfers with the other two antibiotics (*Figure 2—figure supplement 1*), indicating an evolutionary constraint associated with the antibiotic DOR. We can rule out the alternative hypotheses that the reduced growth is explained by a stronger initial reduction in bacterial population size by DOR in comparison to the other two drugs or increased stochastic variation in dosage effects. All treatments were initiated using specifically standardized IC75 dosage (see Materials and methods) and at the IC75, DOR showed very little variation (*Figure 1—figure supplement 1*). We thus hypothesize that the observed evolutionary constraint may be due to lower rate of DOR resistance emergence.

To understand the dynamics of early adaptation in more detail, we measured the resistance profiles of 16 evolved populations after transfers 12 and 48 from the different antibiotic treatments (representing the end of phases 1 and 2, respectively; *Figure 2B, C*, *Figure 2—figure supplements 2–5*, *Supplementary file 1B–F*; see Materials and methods). We randomly sampled 20 bacterial colonies from each population and characterized their resistance profile by broth microdilution. Resistance was measured for the three antibiotics of the evolution experiment and two additional clinically relevant antibiotics from different classes, ciprofloxacin and gentamicin. The resistance profiles in the early and the mid phases were found to be distinctly different. Resistance to the used β-lactams increased across the two time points only in some treatments, but not all (*Figure 2B, C*, *Figure 2—figure supplement 4*, *Figure 2—figure supplement 5*, *Supplementary file 1F*), suggesting treatment-dependent evolutionary responses to the antibiotics. We assessed how the main treatment types varied in their β-lactam resistance using a GLM for each phase separately. Most treatment types varied significantly from each other in their multidrug β-lactam resistance in both phases (*Supplementary file 1C, D*). The multidrug resistance in the early phase was in most cases constrained by the susceptibility to DOR (e.g., in the switching and monotherapy treatments). We additionally observed collateral responses of the treatment to the two non-β-lactams, which increased

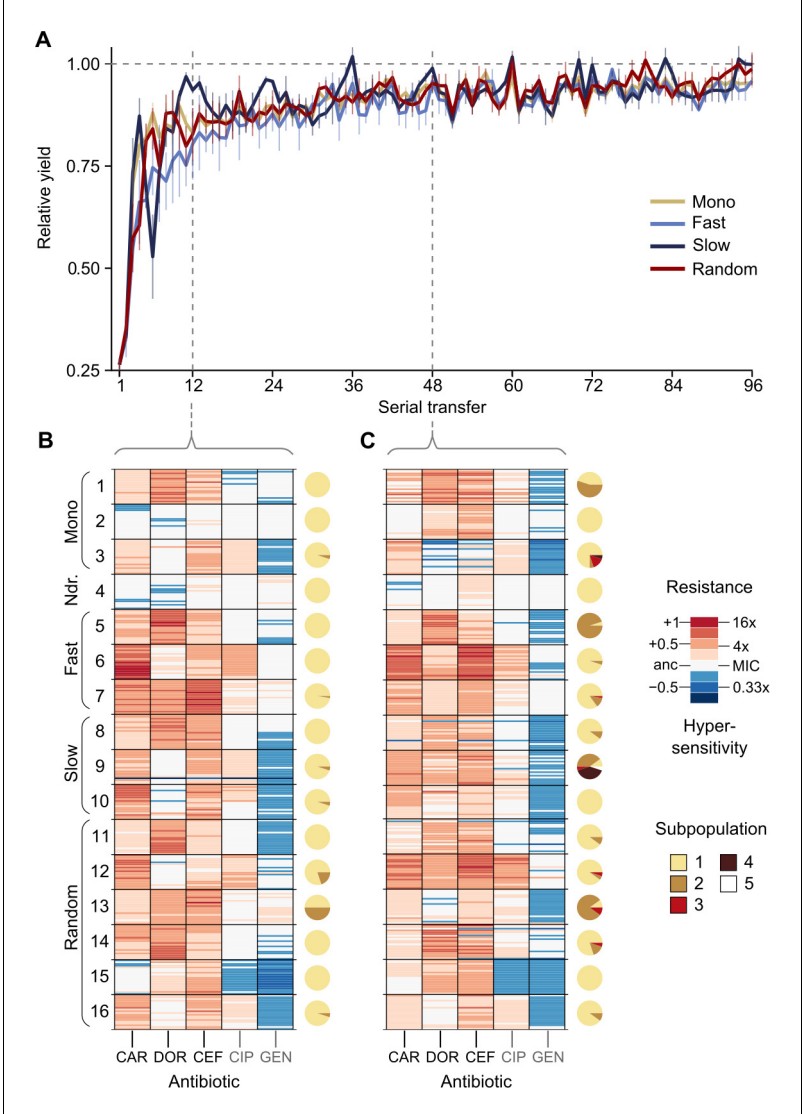

**Figure 2.** Resistance to doripenem is constrained in the CAR-CEF-DOR triple β-lactam experiment. (**A**) Rapid adaptive increase of biomass yields relative to the untreated reference populations (mean ± CI95; n = 3–6 protocols per treatment type and 12 biological replicates per sequence; extinct lineages excluded). Vertical dotted lines separate the three growth phases. Evolved changes in the susceptibility to the treatment antibiotics CAR, DOR, and CEF and the non-treatment antibiotics CIP and GEN after transfer 12 (**B**) or transfer 48 (**C**), evaluated with 20 isolates each for the 16 representative adapting populations at each time point. Mono 1 is monotherapy with CAR, mono 2 is monotherapy with DOR, and mono 3 is monotherapy with CEF. The evolution of resistance and hypersensitivity is indicated by red and blue colors, respectively, given for the considered isolates as horizontal lines (total of 640 isolates), sorted according to evolution treatment (main rows in the figures) and tested antibiotics (main columns; antibiotics given at the bottom). Pie charts on the right show phenotypic within-population diversity, where different colors indicate subpopulations inferred from hierarchical clustering of resistance phenotypes. CAR: carbenicillin; CEF: cefsulodin; CIP: ciprofloxacin; DOR: doripenem; GEN: gentamicin. The following supplementary material is available for Figure 2: *Figure 2—figure supplement 1*, *Figure 2—figure supplement 2*, *Figure 2—figure supplement 3*, *Figure 2—figure supplement 4*, *Figure 2—figure supplement 5*, *Figure 2—source data 1*, *Figure 2—figure supplement 1—source data 1*, *Figure 2—figure supplement 2—source data 1*, *Figure 2—figure supplement 3—source data 1*, *Figure 2—figure supplement 5—source data 1*, *Supplementary file 1B*.

The online version of this article includes the following source data and figure supplement(s) for figure 2:

**Source data 1.** Source data for the panels of *Figure 2*.

**Figure supplement 1.** Growth dynamics in fast sequential protocols.

*Figure 2 continued on next page*

*Figure 2 continued*

**Figure supplement 1—source data 1.** Source data for *Figure 2—figure supplement 1*.

**Figure supplement 2.** Dose-response curve distributions for Exp. 3 with CAR-DOR-CEF, underlying *Figure 2B and C*.

**Figure supplement 2—source data 1.** Source data for *Figure 2—figure supplement 2*.

**Figure supplement 3.** Relation between the resistance values and the fold-change of the minimal inhibitory concentrations (MIC) approximated by IC90.

**Figure supplement 3—source data 1.** Source data for *Figure 2—figure supplement 3*.

**Figure supplement 4.** Change of population antibiotic resistance between transfer 12 (indicated in gray) and transfer 48 (indicated in red).

**Figure supplement 5.** Population multidrug resistance after (**A**) transfer 12 and (**B**) transfer 48.

**Figure supplement 5—source data 1.** Source data for *Figure 2—figure supplement 5*.

over time. We further used hierarchical clustering of the resistance profiles to assess the presence of subpopulations, followed by calculation of Shannon diversity for each population at both transfers. We found population diversity to be significantly higher at transfer 48 as compared to transfer 12 (ANOVA, $F = 6.2060$, p=0.01893, *Supplementary file 1E*), indicating a diversification of the evolving lineages over time. Taken together, the population analysis of resistance profiles indicates that resistance evolution depends on the exact treatment protocol and that the dynamics of resistance emergence to DOR may be key for the observed deceleration of β-lactam adaptation in the fast-regular treatments.

To identify the genomic changes underlying the first steps of β-lactam adaptation, we sequenced 33 whole genomes of the evolved and characterized isolates from the monotherapy, fast-regular, and slow-regular treatment types. Specifically, we sequenced three isolates from each population representing the distinct phenotypic subpopulations, assessed above. We found that all isolates, except those that received DOR monotherapy, had mutations in known resistance genes by the end of the early phase (*Table 1*). This agreed with the inferred resistance profiles where isolates from the

**Table 1.** Evolved genetic changes inferred from whole-genome sequencing.

| Treatment type | ID[*] | AA change[†] | Gene name | Annotation | Freq[‡] |
|---|---|---|---|---|---|
| Monotherapy | 1 | V471G | *ftsI* | Peptidoglycan synthesis | 3/3 |
| | 2[§] | N242S | *ftsI* | Peptidoglycan synthesis | 3/3 |
| | 3 | T157P | *pepA* | Virulence | 3/3 |
| Fast-rgular | 5 | V471G | *ftsI* | Peptidoglycan synthesis | 3/3 |
| | 6 | K26 | *nalD* | Efflux | 3/3 |
| | | S379ISR | *rmcA* | Biofilm maintenance | 1/3 |
| | 7 | R220C | *phoQ* | Two-component | 3/3 |
| | | - | PA14_55631 | 23srRNA, translation | 1/3 |
| Slow-rgular | 8 | V471G | *ftsI* | Peptidoglycan synthesis | 3/3 |
| | 9 | D357N | *pepA* | Virulence | 3/3 |
| | 10 | T157P | *pepA* | Virulence | 3/3 |
| | | E115VAAWIPK | PA14_21540 | Lipid metabolism (3-exoacyl ACP synthase) | 1/3 |
| | | Q117AEEQ | PA14_21540 | Lipid metabolism (3-exoacyl ACP synthase) | 1/3 |
| | | R178C | *zipA* | Cell division | 2/3 |
| | | P483PEP | *dnaX* | Cell division | 1/3 |

[*] Individual treatment of evolution experiment.

[†] Amino acid change.

[‡] Occurrence frequency of the identified variant (before slash) out of the total number of isolates sequenced (behind slash).

[§] Mutations listed are from isolates obtained from the populations frozen at transfer 48, no variants were found in the isolates from transfer 12.

The online version of this article includes the following source data for Table 1:

Source data 1. Source data for the summary of the genome sequencing analysis shown in *Table 1*.

DOR monotherapy did not show a noticeable amount of resistance at that stage (*Figure 2B*). DOR resistance was, however, found at the end of the middle phase (*Figure 2C*), and this was mirrored in the genomics with a non-synonymous mutation in the gene *ftsI*. This gene codes for the penicillin binding protein 3 (PBP3) (*Liao and Hancock, 1995*), a common target of the three β-lactams (*Davies et al., 2008*; *Fontana et al., 2000*; *Rodriguez-Tebár et al., 1982*; *Rodríguez-Tebar et al., 1982*; *Zimmermann, 1980*). *ftsI* was also found to be mutated in isolates from CAR monotherapy, although at a different site within the gene and associated with a different resistance profile than the DOR-associated *ftsI* variant (*Figure 2B*). Isolates from CEF monotherapy contained mutations in *pepA*. This gene is responsible for the production of a protein required for cytotoxicity and virulence in *P. aeruginosa* (*Hauser et al., 1998*). Although its role in antimicrobial resistance remains to be studied in detail, it was previously found to be mutated in *P. aeruginosa* strains resistant to certain β-lactams (*Cabot et al., 2018*; *Sanz-García et al., 2018*). The switching treatments selected for mutations in the above-listed and also in some additional genes. In particular, we identified mutations in *nalD* and *phoQ*, a negative regulator of the MexAB-OprM efflux pump and a two-component system, respectively. Mutations in these genes account for resistance to a variety of drugs in *P. aeruginosa* (*Barbosa et al., 2021*; *Sobel et al., 2005*). Further mutations were identified in some non-canonical β-lactam resistance genes such as *rmcA*, 23srRNA, *3-oxoacyl synthase*, *dnaX*, and *zipA* (*Table 1*). Taken together, mutations in both canonical and non-canonical targets of β-lactam selection were identified in our experiment, and among these, DOR resistance mutations were found only later in the experiment, consistent with the obtained resistance profiles (*Figure 2B, C*).

Based on our detailed characterization of the CAR-CEF-DOR triple β-lactam experiment, we conclude that DOR has a key role in restricting evolutionary rescue as evidenced by the delayed acquisition of genetic resistance to it.

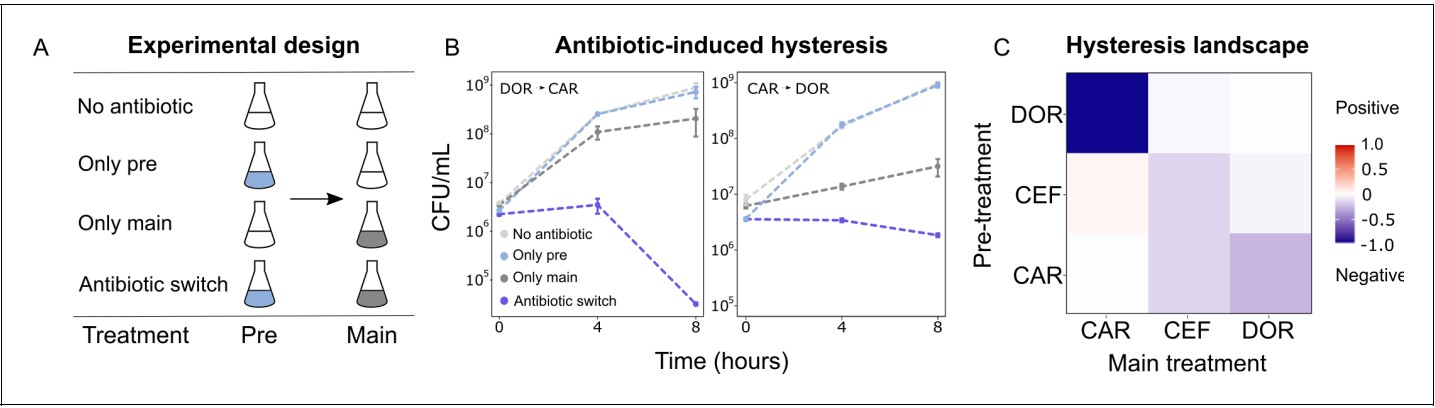

**Figure 3.** Negative hysteresis is common among the tested β-lactam antibiotics. (**A**) Hysteresis effects were measured using the previously established experimental approach (see Materials and methods). (**B**) Bacterial counts were plotted over time after the pretreatment to obtain time-kill curves (mean ± sem, n = 3). Level of hysteresis was quantified as the difference between the antibiotic switch and the only main curves. Negative values indicate negative hysteresis and positive values indicate positive hysteresis. (**C**) Heatmap of hysteresis levels between all nine combinations of the three β-lactams. DOR and CAR show asymmetric bidirectional negative hysteresis. Negative hysteresis is also observed in switches from CEF to CEF and CAR to CEF. Weak positive hysteresis is found for the switch from CEF to CAR. The following supplementary material is available for Figure 3: *Figure 3—figure supplement 1*, *Figure 3—figure supplement 2*, *Figure 3—source data 1*, *Figure 3—figure supplement 1—source data 1*. CAR: carbenicillin; CEF: cefsulodin; DOR: doripenem.

The online version of this article includes the following source data and figure supplement(s) for figure 3:

**Source data 1.** Source data for the panels of *Figure 3*.

**Figure supplement 1.** Time-kill curves of hysteresis experiments for the combinations not presented in *Figure 3* (mean ± sem, n = 3).

**Figure supplement 1—source data 1.** Source data for *Figure 3—figure supplement 1*.

**Figure supplement 2.** Hysteresis effects quantified as area under the curve (AUC) difference between the 'only main' and 'antibiotic switch' curves from the time-kill dynamics.

## Asymmetric bidirectional hysteresis was identified between doripenem and carbenicillin

As extinction was associated with antibiotic switches, we next focused on selective events that can occur at drug switches, such as hysteresis, an inducible physiological change. We characterized the complete hysteresis landscape between the three β-lactams: CAR, DOR, and CEF. We pretreated exponential phase cells with an antibiotic for only 15 min to ensure that cells are physiologically challenged but not subject to differential killing or replication. The pretreatment was followed by a change to fresh medium containing a second antibiotic as main treatment. We included controls of no pretreatment, or no main treatment (*Figure 3A*). We found that negative hysteresis existed for several switches between the β-lactams (*Figure 3B, C*, *Figure 3—figure supplement 1*, *Figure 3—figure supplement 2*). DOR and CAR displayed asymmetric bidirectional negative hysteresis with the switch from DOR to CAR, resulting in stronger negative hysteresis than the reverse. Negative hysteresis was also observed in the switch from CAR to CEF and CEF to CEF. To our surprise, only a single case of weak positive hysteresis was observed, although we generally anticipated it given that *P. aeruginosa* produces the AmpC β-lactamase (*Livermore, 1995*). We conclude that negative hysteresis is abundant between the studied β-lactams and is a potential predictor of treatment potency in the sequential β-lactam treatments.

## Probability of direct and indirect resistance was the least for doripenem

Since resistance to DOR was constrained in both the monotherapy and the switching treatments (*Figure 2B*), we hypothesized that DOR resistance was difficult to achieve compared to the other two β-lactams. Resistance against a given drug can arise because of spontaneous direct resistance and/or because of collateral resistance from the preceding antibiotics in the sequence. As a first step, we thus measured the spontaneous direct resistance rate with the classic fluctuation assay using identical inhibitory concentrations of the three antibiotics (*Luria and Delbrück, 1943*; *Figure 4A*, *Supplementary file 1G*). To determine the probability of indirect resistance in a second step, we isolated the obtained single-step mutants and quantified the fraction of cross-resistance towards the other two β-lactams with a patching assay (*Figure 4A*). We used a comparatively large number of spontaneous mutants for this analysis (n = 60 per antibiotic) to capture the stochastic nature of evolution and, in this context, the potential importance of collateral effects for bacterial adaptation, as previously emphasized (*Nichol et al., 2019*). We found that the spontaneous resistance rate was significantly lower for DOR than for CAR and CEF (likelihood ratio test, p<0.0001 and p<0.01, respectively; *Supplementary file 1H*, *Figure 4B*). Moreover, the resulting cross-resistance effects (*Figure 4C*) were particularly common towards CAR (93% of clones with spontaneous CEF resistance and 71% with DOR resistance) and CEF (73% of originally CAR-resistant clones and 67% DOR-resistant clones). By contrast, the smallest levels of cross-resistance were expressed towards DOR (36% of originally CAR-resistant clones and 50% CEF-resistant clones). The overall fraction of cross-resistant clones was significantly smaller towards DOR than either CEF or CAR (Fisher's exact test, p<0.0004; *Supplementary file 1I*). We conclude that of the three β-lactams DOR had the lowest probability for both direct and indirect resistance, thereby providing experimental support to the indication of constrained DOR resistance evolution obtained from the detailed phenotypic and genomic characterization of the evolved bacteria (*Figure 2*, *Table 1*).

## The rate of spontaneous resistance and resulting cross-resistance determine treatment efficacy

We used the collected information to identify the critical determinant(s) of treatment efficacy in the CAR-CEF-DOR triple β-lactam experiment. We assessed the influence of either the two experimental predictors (switching rate, temporal irregularity) or the three biological predictors (hysteresis, probability of spontaneous resistance, and resulting cross-resistance) on each of the evolutionary responses extinction, rate of growth adaptation, and multidrug resistance, using separate GLM-based analyses (see Materials and methods; *Supplementary file 1J–O*). For the biological predictors, we calculated the levels of cumulative hysteresis, cumulative probability of spontaneous resistance, and the cumulative levels of cross-resistance in each of the 16 individual treatments up to transfer 12 (see Materials and methods). We focused our analysis on the early phase of evolution up

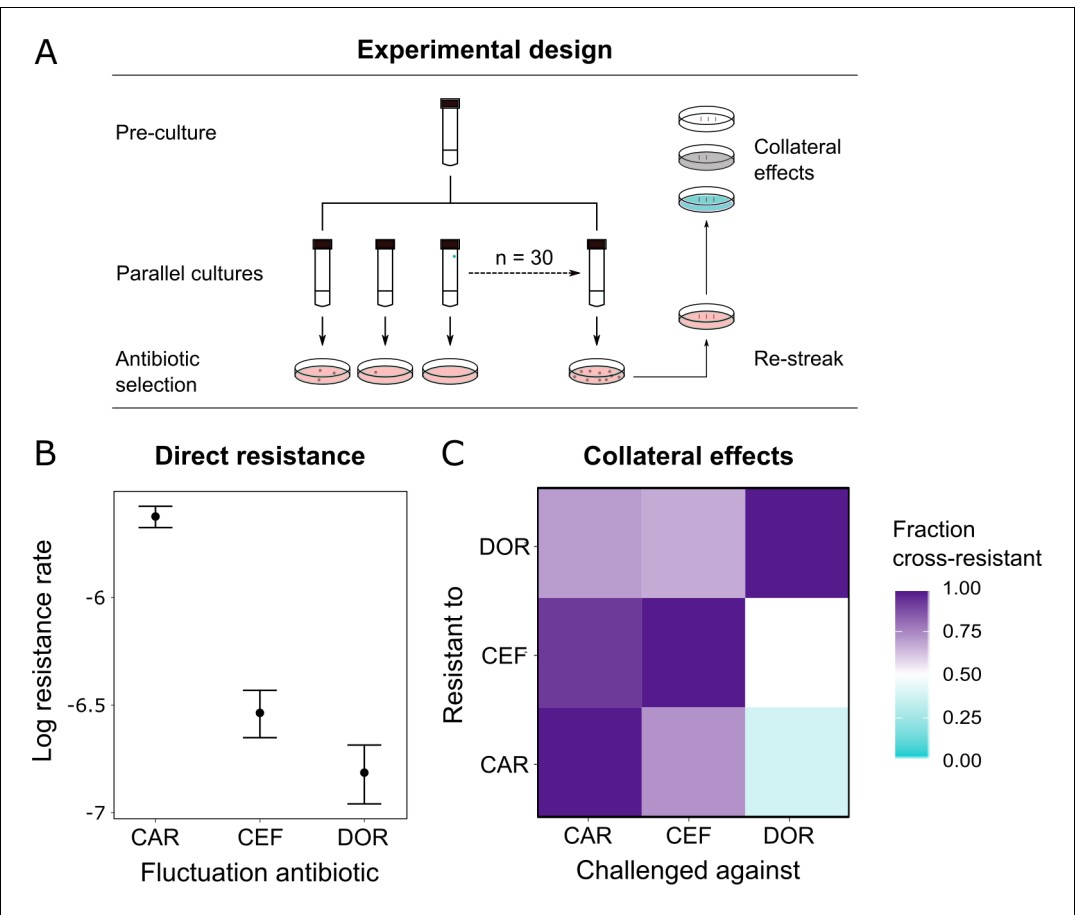

**Figure 4.** Doripenem has the lowest rates of direct and indirect resistance. (A) Schematic of the experimental protocol to determine spontaneous rates of resistance on each of the three β-lactams and the resulting collateral landscape. Briefly, an overnight culture was taken and split into 30 parallel cultures where bacteria were allowed to divide in the absence of an antibiotic and any other constraint. Spontaneous resistant mutants were selected on minimal inhibitory concentration (MIC) plates and restreaked to ensure genetic resistance. These mutants were then patched on MIC plates of the other two β-lactams to test for cross-resistance. (B) Comparison of rates of spontaneous resistance on the three β-lactams on a Log10 scale. Error bars depict CI95. All comparisons were found to be significantly different from each other (likelihood ratio test; CAR vs. CEF p<0.0001, CAR vs. DOR p<0.0001, and DOR vs. CEF p<0.01). (C) Landscape of collateral effects between the three β-lactams. Fraction of cross-resistant mutants per antibiotic combination is plotted. DOR has the least cases of cross-resistance of the three. A total of 60 mutants per antibiotic were used for collateral effect testing. The following supplementary material is available for Figure 4: *Figure 4—source data 1*, *Supplementary file 1G–I*. CAR: carbenicillin; CEF: cefsulodin; DOR: doripenem.

The online version of this article includes the following source data for figure 4:

**Source data 1.** Source data for *Figure 4*.

---

to transfer 12 as it appeared most critical for treatment efficacy, especially for population extinctions that usually occurred early (*Figure 1E*). Our analysis revealed that extinction was significantly associated with both the experimental predictors, switching rate (GLM, $F$ = 14.44, p=0.0042, *Figure 5B*, *Supplementary file 1J–M*) and temporal irregularity (GLM, $F$ = 10.53, p=0.0101, *Supplementary file 1M*). Temporal irregularity further showed a statistical trend with multidrug resistance (GLM, $F$ = 4.19, p=0.0711, *Supplementary file 1M*). From our biological predictors, the cumulative cross-resistant fraction showed a significant association with extinction (GLM, $F$ = 10.42, p=0.0121, *Supplementary file 1O*), while cumulative probability of spontaneous resistance showed a statistical trend (GLM, $F$ = 4.14, p=0.0763, *Supplementary file 1O*). Indeed, the cumulative cross-resistant fraction and also the cumulative probability of spontaneous resistance are strongly

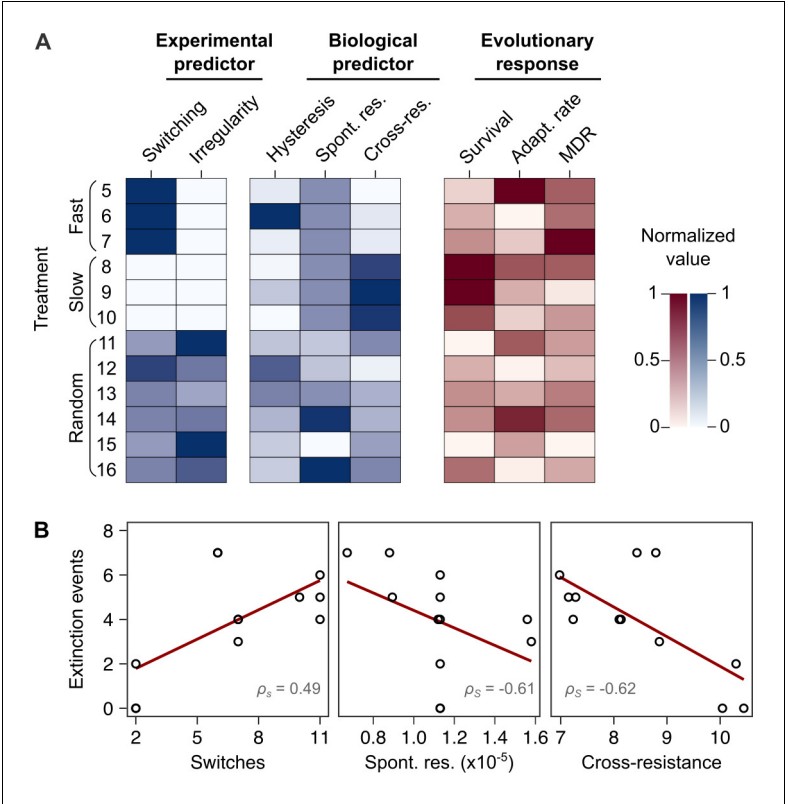

**Figure 5.** Bacterial extinction is correlated to switching rate, spontaneous rate of resistance, and spontaneous cross-resistance. (**A**) Variation in experimental parameters, potential biological predictors, and the measured traits up to transfer 12. The experimental parameters include switching rate and regularity of change (high irregularity in dark). Potential biological predictors are cumulative levels of hysteresis (dark indicates protective effects), cumulative probabilities of spontaneous resistance (Spont. res., dark indicates higher probability), and cumulative level of collateral effects (Cross-res., dark indicates high fraction of cross-resistance). The evolutionary response was measured for population survival (max = 12), adaptation rate (Adapt. rate, $n \leq 12$, extinct lineages excluded), evolved multidrug resistance (MDR) to treatment antibiotics CAR, DOR, and CEF (MDR, n = 16). (**B**) Variation in extinction was best explained by collateral effects between the antibiotics (for illustrative purposes, the red line depicts linear regression and $\rho_S$ the Spearman's rank correlation coefficient). The following supplementary material is available for Figure 5: *Figure 5—figure supplement 1*, *Figure 5—source data 1*, *Supplementary file 1J–O*. CAR: carbenicillin; CEF: cefsulodin; DOR: doripenem.

The online version of this article includes the following source data and figure supplement(s) for figure 5:

**Source data 1.** Source data for *Figure 5*.

**Figure supplement 1.** Correlation of switching rate with cumulative levels of collateral effects.

correlated with extinction (*Figure 5B*). The cumulative cross-resistant fraction is also strongly correlated with switching rate (*Figure 5—figure supplement 1*), most likely explaining the latter impact on extinction. By contrast, cumulative hysteresis levels did not have a significant influence on any of the evolutionary responses (GLM, $F = 0.16$, p=0.7015, *Supplementary file 1O*). Taken together, our results suggest that in our sequential CAR-CEF-DOR treatments the switching rate, temporal irregularity of antibiotics, the probability of spontaneous resistance, and especially the resulting collateral effects (maximized by switching rate) determine treatment efficacy through their effect on bacterial extinction. The limiting factor appears to be constrained evolution of resistance and low levels of cross-resistance to DOR.

## Discussion

Treatment with multiple β-lactam antibiotics is generally avoided due to the perceived fear of therapy failure from cross-resistance. Our work now challenges this widespread belief. We characterized the ability of replicate *P. aeruginosa* populations to evolve de novo resistance to sequential treatments with different drug sets. To our surprise, we found that sets of three β-lactams constrained bacterial adaptation by reducing bacterial survival. We demonstrate that treatment potency was determined by variation in the spontaneous rate of resistance to the β-lactams and the resulting collateral effects across sequential treatment protocols.

Our initial screen of sequential protocols with different antibiotic triplets revealed that the triple β-lactam sequences are at least as effective at causing extinction as sequences of antibiotics with distinct modes of actions. This finding is at first sight counterintuitive, but at second sight not completely unexpected. The joint application of two β-lactam drugs was in fact tested and found effective in a few previous studies (*Rahme et al., 2014*). For example, the β-lactam aztreonam was shown to interact synergistically with four other β-lactam drugs against multiple resistant isolates of *Enterobacteriaceae* and *P. aeruginosa* in vitro (*Buesing and Jorgensen, 1984*). A combination of ticarcillin with ceftazidime produced high efficacy in a rat peritonitis model (*Shyu et al., 1987*). In a treatment of bacterial soft tissue infections, the combination of cefotaxime and mecillinam led to higher clinical response rates than the tested monotherapy (*File and Tan, 1983*). Further, the dual β-lactam combination of ceftazidime plus piperacillin was as effective as the combination of ceftazidime and tobramycin in granulocytopenic cancer patients (*Joshi et al., 1993*). More recent studies demonstrated that a triple combination of meropenem, piperacillin, and tazobactam successfully constrained resistance evolution in Methicillin-resistant *Staphylococcus aureus* (MRSA), both in vitro and in a mouse model (*Gonzales et al., 2015*). In addition, the combination of cefotaxime and mecillinam was effective against *Salmonella enterica* harboring a mutant β-lactamase in a mouse model (*Rosenkilde et al., 2019*). Our findings add to the high potency of treatments with multiple β-lactams. We conclude that the use of multiple β-lactams, either as a combination or sequentially, is a commonly underappreciated form of therapy and its use opens new avenues to better utilize our existing antibiotic armamentarium.

Spontaneous rate of antibiotic resistance was found to play a critical role in the success of the CAR-CEF-DOR sequential treatment. The probability of spontaneous resistance on all three β-lactams was significantly different, with the rate of DOR resistance being the lowest. These rates determined the overall probability of acquiring direct resistance in treatment, which significantly correlated with the frequency of population extinction (*Figure 5B*). Resistance rates were previously shown to vary towards different antibiotics, for example, in *Escherichia coli* (*El Meouche and Dunlop, 2018*) and *P. aeruginosa* (*Oliver et al., 2004*). This variation can arise from genetic factors such as mutational target space and physiological factors like activation of the bacterial SOS response (*Martinez and Baquero, 2000*). Such information on resistance rates has so far been used for predicting the occurrence of resistance against single drugs, and antibiotics that target multiple pathways in a cell are considered advantageous in this context (*Ross-Gillespie and Kümmerli, 2014*). One example of the latter are compounds against *S. aureus* that inhibit both DNA gyrase and topoisomerase IV (*Nyerges et al., 2020*). The rate of resistance emergence may also be reduced by using adjuvants that target the SOS response (*Bell and MacLean, 2018*), as previously shown for compounds interfering with LexA activity leading to reduced resistance rates to ciprofloxacin and rifampicin in *E. coli* (*Cirz et al., 2005*). Our study extends the role of resistance rates of antibiotics beyond this convention. We show that inclusion of an antibiotic with relatively low spontaneous resistance emergence can enhance the potency of a sequential treatment design.

What could be the underlying reasons for the particular importance of DOR compared with the other β-lactams? DOR belongs to the carbapenem subclass of the β-lactam antibiotics. Carbapenems possess broad activity against Gram-positive and Gram-negative bacteria (*Papp-Wallace et al., 2011*) and are active against many β-lactamase-producing microbes since their thiazolidinic ring makes them relatively resistant to β-lactamase-mediated hydrolysis (*Schafer et al., 2009*). In contrast, the penicillin CAR is active mostly (albeit not exclusively) against Gram-negative bacteria (*Castle, 2007*) while the activity of the cephalosporin CEF is restricted to *P. aeruginosa* (*Wright, 1986*). Within *P. aeruginosa*, all three antibiotics show high potency against a large variety of clinical isolates (*Castanheira et al., 2009*; *Neu and Scully, 1984*; *Traub and Raymond, 1970*).

Resistance rates for β-lactam antibiotics were assessed with different approaches across *P. aeruginosa* strains and clinical isolates, consistently showing that DOR has a particularly low propensity to select for resistance mutations, even when compared to other carbapenems (*Barbosa et al., 2017*; *Barbosa et al., 2021*; *Sakyo et al., 2006*; *Tanimoto et al., 2008*; *Mushtaq et al., 2004*; *Fujimura et al., 2009*). Therefore, the phenotype of reduced spontaneous resistance to DOR appears to be robustly expressed across different *P. aeruginosa* genotypes and does not extend to other carbapenems or β-lactams. One possible reason for this pattern may be variation in the range of β-lactam target proteins, in this case the penicillin binding proteins (PBPs), and where DOR is known to bind more of these PBPs than do CAR or CEF (*Davies et al., 2008*; *Fontana et al., 2000*; *Rodriguez-Tebár et al., 1982*; *Rodríguez-Tebar et al., 1982*; *Zapun et al., 2008*; *Zimmermann, 1980*). Thus, target resistance to DOR would likely require a larger number of mutations than that to other β-lactams. Interestingly, another carbapenem, meropenem, targets the same PBPs as DOR (*Davies et al., 2008*) but has a higher resistance rate, suggesting that the underlying reasons for resistance rate variation are multifactorial. Taken together, effective resistance mutations against DOR seem to be less commonly available in *P. aeruginosa* in comparison to that against other drugs, including the here used CEF and CAR.

A key determinant of treatment potency was the reduced level of spontaneous cross-resistance to the sequentially applied drugs (*Figure 5B*). This effect was maximized by the switching rate (*Figure 5—figure supplement 1*). Our findings are consistent with the previously and repeatedly proposed importance of collateral sensitivity for the efficacy of sequential treatment protocols (*Barbosa et al., 2019*; *Hernando-Amado et al., 2020*; *Imamovic and Sommer, 2013*; *Kim et al., 2014*; *Maltas and Wood, 2019*; *Yen and Papin, 2017*). Even though we did not measure collateral sensitivity directly, the lack of cross-resistance is related as it indicates that the mutant cells, which have become resistant to one drug, maintain at least ancestral levels of susceptibility against the second drug. Moreover, our study focused on spontaneous emergence of cross-resistance (or lack thereof). By contrast, many previous studies established collateral effects after bacteria evolved resistance to the first drug over many generations, often followed by only a single antibiotic switch to assess the impact of collateral sensitivity on therapy success (*Barbosa et al., 2019*; *Hernando-Amado et al., 2020*; *Imamovic and Sommer, 2013*; *Yen and Papin, 2017*). Surprisingly, our study revealed potentially beneficial collateral effects between antibiotics of the same class. In fact, we chose these three β-lactams because our previous work demonstrated cross-resistance between most of them, although inferred upon multigenerational adaptation to the first drug (*Barbosa et al., 2017*). Our current finding of a lack of cross-resistance among some of these drugs now suggests that spontaneous mutants may have different collateral profiles than the lines, which adapted over many generations. Our results further suggest that the collateral effects of spontaneous mutants are likely to be more pertinent for the design of sequential treatments with fast switches among antibiotics. This suggestion is supported by two previous studies, in which the efficacy of fast sequential treatments was optimized by considering collateral effects for either single-step mutants of *S. aureus*, obtained after 20 hr exposure to three distinct antibiotics for 20 hr (*Kim et al., 2014*), or from *Enterococcus faecalis* populations adapted over 2 days to four distinct antibiotics (*Maltas and Wood, 2019*). As a side note, it is particularly interesting that our detailed resistance analysis consistently revealed almost all treatments to cause the evolution of collateral sensitivity towards the aminoglycoside gentamicin, but not the fluoroquinolone ciprofloxacin (*Figure 2B, C*), possibly indicating yet another treatment option – in cases where the applied triple β-lactam sequential protocols fail.

Temporal irregularity was additionally found to constrain bacterial adaptation. When bacteria experienced the antibiotics in an irregular pattern, this caused significantly increased extinction and to some degree reduced multidrug resistance. With CAR-DOR-CEF, the lowest multidrug resistance was observed in random sequential treatments (*Figure 2—figure supplement 5*), as also previously observed with CIP-GEN-CAR (*Roemhild et al., 2018*). Environmental change anticipation has been documented in several microorganisms (*Mitchell et al., 2009*; *Mitchell and Pilpel, 2011*), indicating their capability to specifically adapt to regular environmental change. Stochastic changes can make it harder to evolve anticipation (*Roemhild and Schulenburg, 2019*). Stochastic changes in environmental parameters were indeed found to constrain fitness in evolving bacteria (*Hughes et al., 2007*) and viruses (*Alto et al., 2013*). We show that irregular antibiotic sequences have potential to inhibit bacterial resistance evolution.

Unexpectedly, we further identified negative hysteresis for multiple combinations of the three β-lactams. However, cumulative hysteresis levels per treatment did not significantly associate with any of our measured evolutionary responses. In our previous study (*Roemhild et al., 2018*), within the CAR-CIP-GEN combination, negative hysteresis was expressed for the switches from CAR to GEN and CIP to GEN. Yet, only the CAR-GEN hysteresis was significantly associated to the evolutionary responses. Thus, hysteresis interactions can exist between antibiotics from the same or different classes, but they need not impact the evolutionary outcome of a sequential treatment protocol each time. In the current study, it appears that spontaneous resistance effects and the resulting cross-resistance effects are dominant over the β-lactam hysteresis. One potential explanation could be that insensitivity to β-lactam hysteresis evolves quickly. Nevertheless, it clearly warrants further research to assess whether negative hysteresis between the β-lactam drugs is robustly shown across strains of *P. aeruginosa* or other bacterial species and can somehow be exploited in sequential therapy, in analogy to the previous results with antibiotics from different classes (*Roemhild et al., 2018*).

Taken together, our study highlights that the available antibiotics offer unexplored, highly potent treatment options that can be harnessed to counter the spread of drug resistance. It further underscores the importance of evolutionary trade-offs such as reduced cross-resistance in treatment design and introduces spontaneous resistance rates of component antibiotics as a guiding principle for sequential treatments. It is ironic that the differential cross-resistance landscape of the β-lactams was a key factor contributing to treatment potency, even though the risk of cross-resistance is usually used to reject β-lactam-exclusive treatments. The underlying reasons for differential spontaneous and long-term cross-resistance between these drugs (including the underlying molecular mechanisms) are as yet unknown and clearly deserve further attention in the future. We conclude that a detailed understanding of both spontaneous resistance rates and resulting cross-resistances against different antibiotics should be of particular value to further improve the potency of sequential protocols.

# Materials and methods

**Key resources table**

| Reagent type (species) or resource | Designation | Source or reference | Identifiers | Additional information |
|---|---|---|---|---|
| Strain, strain background (*Pseudomonas aeruginosa*) | PA14 | https://doi.org/10.1126/science.7604262 | UCBPP-PA14 | |
| Chemical compound, drug | AZL (azlocillin) | Sigma | A7926-1G | |
| Chemical compound, drug | CAR (carbenicillin) | Carl Roth | 6344.2 | |
| Chemical compound, drug | CIP (ciprofloxacin) | Sigma | 17850-5 G-F | |
| Chemical compound, drug | CEF (cefsulodin) | Carl Roth | 4014.2 | |
| Chemical compound, drug | CEZ (ceftazidime) | Sigma | C3809.1G | |
| Chemical compound, drug | DOR (doripenem) | Sigma | 32138-25 MG | |
| Chemical compound, drug | GEN (gentamicin) | Carl Roth | 2475.1 | |

*Continued on next page*

*Continued*

| Reagent type (species) or resource | Designation | Source or reference | Identifiers | Additional information |
|---|---|---|---|---|
| Chemical compound, drug | STR (streptomycin) | Sigma | S6501-5 | |
| Chemical compound, drug | TIC (ticarcillin) | Sigma | T5639-1G | |
| Software, algorithm | R: A language and environment for statistical computing | https://www.R-project.org/ | | |

## Materials

All experiments were performed with *P. aeruginosa* UCBPP-PA14 (*Rahme et al., 1995*). Bacteria were grown in M9 minimal medium supplemented with glucose (2 g/L), citrate (0.58 g/L), and casamino acids (1 g/L) or on M9 minimal agar (1.5%) or Lysogeny broth (LB) agar. Antibiotics were added as indicated. Cultures and plates were incubated at 37°C. Experiments included biological replicates (initiated with independent clones of the bacteria, which were grown separately before the start of the experiment, or independent evolutionary lineages from the respective evolution treatments) and technical replicates (initiated from the same starting culture of the bacteria), as indicated below. For the experiments, treatment groups were run in parallel and randomized. Treatment names were masked in order to minimize observer bias.

## Dose-response curves of ancestor

We used dose-response curves based on broth microdilution in order to determine antibiotic concentration causing inhibition level of 25% growth yield relative of untreated controls (inhibitory concentration 75 [IC75]) for the antibiotics azlocillin (AZL), carbenicillin (CAR), ciprofloxacin (CIP), cefsulodin (CEF), ceftazidime (CTZ), doripenem (DOR), gentamicin (GEN), and ticarcillin (TIC; see *Supplementary file 1A* for details on antibiotics). Briefly, bacteria were grown to exponential phase ($OD_{600}$ = 0.08) and inoculated into 96-well plates (100 µL per well, $5 \times 10^6$ CFU/mL) containing linear concentration ranges close to MIC of the antibiotics in M9 medium. Antibiotic concentrations were randomized spatially. Bacteria were incubated for 12 hr after which optical density was measured in BioTek EON plate readers at 600 nm ($OD_{600}$). We included six biological replicates and 1–2 technical replicates per concentration and antibiotic. Optical density was plotted against antibiotic concentration to obtain a dose-response curve. Model fitting was carried out using the package *drc* (*Ritz et al., 2015*) in the statistical environment R (*R Development Core Team, 2020*) and the fitted curve was used to predict IC75 values (*Figure 1—figure supplement 1*).

## Evolution experiments

We carried out evolution experiments with the various combination of antibiotics according to the design described previously (*Roemhild et al., 2018*). A total of 16 treatments were included (*Figure 1B*). Treatments 1–4 were constant environments consisting of the monotherapy (#1–3) and no drug control (#4). Treatments 5–10 were the regular switching treatments. They switched between the antibiotics in a regular predictable fashion, either every transfer (fast; #5–7) or every fourth transfer (slow; #8–10). Treatments 11–16 consisted of the random treatments that switched fast in a temporally irregular fashion. The setup was designed to test the effect of switching rate and temporal irregularity.

Every treatment consisted of 12 replicate populations (initiated from six biological replicates × two technical replicates). All populations were started with an inoculum of $5 \times 10^5$ cells. Populations were propagated as 100 µL batch cultures in 96-well plates, with a transfer to fresh medium every 12 hr (transfer size 2% v/v). Antibiotic selection was applied at IC75 throughout. We monitored growth by $OD_{600}$ measurements taken every 15 min through the entire evolution experiment (BioTek Instruments, USA; EON; 37°C, 180 rpm double-orbital shaking). Evolutionary growth dynamics were

assessed by plotting the final OD achieved in every transfer (relative to final OD of no drug control; relative yield). Adaptation rate was calculated with a sliding window approach, where adaptation rate was the inverse of the transfer at which the mean relative yield of a sliding window of 12 transfers reached 0.75 for the first time. Cases of extinction were determined at the end of the experiment by counting wells in which no growth was observed after an additional incubation in antibiotic-free medium. Samples of the populations were frozen in regular intervals in 10% (v/v) DMSO and stored at −80°C for later analysis. The evolution experiments were carried out for a total of 96 transfers.

## Resistance measurements of evolved populations

We characterized populations frozen at transfers 12 and 48 in detail because they represented the early and late phases of the evolution experiment. One population originating from a single biological replicate was chosen per treatment and plated onto LB agar. After incubation at 37°C, 20 colonies from each population were picked randomly and frozen in 10% (v/v) DMSO and stored at −80°C. These colonies, termed isolates, were considered to be representative biological replicates for each population. We constructed dose-response curves for the isolates using for each evolved population one technical replicate per isolate and four technical replicates of the ancestral PA14 strain, as described above, for the antibiotics CAR, CEF, DOR, GEN, and CIP. The integral of this curve for every isolate was calculated and the integral of the ancestral PA14 control subtracted. The resulting value was resistance of the isolate on the said antibiotic. We identified subpopulations in any given population by hierarchical clustering of the resistance profiles, as previously described (*Roemhild et al., 2018*). Resistance of a population was calculated by averaging the resistance of the isolates. Resistance of the population on CAR, CEF, and DOR was added to obtain a single value for multidrug resistance.

## Whole-genome sequencing

From the frozen isolates at transfer 12, we chose three isolates per population (i.e., three biological replicates per population) for whole-genome sequencing to determine possible targets of selection. Each resistance cluster in the population was represented in the sequenced isolates. For the DOR monotherapy, isolates from transfer 48 were also sequenced as no phenotypic resistance was observed at transfer 12. Frozen isolates were thawed and grown in M9 medium at 37°C for 16–20 hr. We extracted DNA using a modified CTAB protocol (*von der Schulenburg et al., 2001*) and sequenced it at the Competence Centre for Genomic Analysis Kiel (CCGA Kiel; Institute for Clinical Microbiology, University Hospital Kiel), using Illumina Nextera DNA Flex library preparation and the MiSeq paired-end technology (2 × 300 bp). Quality control on the resulting raw reads was performed with FastQC (*Andrews, 2010*) and low-quality reads were trimmed using Trimmomatic (*Bolger et al., 2014*). We then used MarkDuplicates from the Picard Toolkit (http://broadinstitute.github.io/picard/) to remove duplicate reads and mapped the remaining reads to the *P. aeruginosa* UCBPP-PA14 genome (available at http://pseudomonas.com/strain/download) using Bowtie2 and samtools (*Langmead and Salzberg, 2012*; *Li et al., 2009*). Variant calling was done using the GATK suite (*Poplin et al., 2018*) and the called variants were annotated using SnpEFF (*Cingolani et al., 2012*) and the Pseudomonas Genome Database (https://www.pseudomonas.com/). We removed all variants that were detected in the no drug control as they likely represent adaptation to the medium and not the antibiotic. The fasta files of all sequenced isolates are available from NCBI under the BioProject number: PRJNA704789.

## Hysteresis testing

The presence of cellular hysteresis was tested, following the previously developed protocol (*Roemhild et al., 2018*). Bacterial cells were grown to exponential phase ($OD_{600}$ = 0.08), diluted 10-fold, and treated with IC75 of the first antibiotic. In the treatments where the pretreatment did not require an antibiotic, none was added. These cells were allowed to incubate for 15 min at 37°C and 150 rpm (pretreatment). After this, the first antibiotic was removed by centrifugation and fresh medium containing IC75 of a second antibiotic was added. In cases where the main treatment did not require an antibiotic, fresh medium without an antibiotic was added. Bacteria were now incubated for 8 hr at 37°C and 150 rpm (main treatment). Bacterial count was monitored through the

main treatment by spotting assays. We used three biological replicates per treatment and, for CFU counting, four technical replicates per biological replicate and treatment. $\log_{10}$ CFU/mL were plotted against time to obtain time-kill curves (*Figure 3B*). The level of hysteresis was calculated as the difference between the antibiotic switch and only main treatment curves.

## Agar dilution

We determined the MIC on M9 agar for the antibiotics CAR, CEF, and DOR according to the EUCAST protocol (https://doi.org/10.1046/j.1469-0691.2000.00142.x) that was modified to account for inoculum effect in our fluctuation assay setup. UCBPP-PA14 was grown in M9 medium at 37°C for 20 hr. $5 \times 10^5$ cells were taken from the stationary phase cultures and spread on M9 agar plates containing doubling dilutions of the antibiotic. Plates were incubated at 37°C for 20–24 hr. MIC was read as the lowest concentration at which no growth of bacteria was seen. MIC determination for each antibiotic was done for three biological replicates (no additional technical replication).

## Fluctuation assay

We measured resistance rates on the three β-lactams using the classic fluctuation assay (*Luria and Delbrück, 1943*). Briefly, a single colony of UCBPP-PA14 was inoculated to 10 mL M9 and incubated at 37°C, 150 rpm for 20 hr. This primary culture was used to start 30 parallel cultures all having a starting concentration of $10^2$ CFU/mL. The parallel cultures were considered biological replicates and incubated at 37°C, 150 rpm for 20 hr. Thereafter, $5 \times 10^5$ cells were plated onto MIC plates of CAR, CEF, and DOR. The plates were incubated for 40 hr at 37°C. The resulting mutant colonies were taken and patched on identical antibiotic MIC plates to ensure genetic resistance. Colonies that grew after patching were counted. We used counts from all 30 cultures to estimate resistance rate on each antibiotic using the package *rSalvador* (*Zheng, 2017*) in R (*R Development Core Team, 2020*).

## Patching assay

We assessed the extent of cross-resistance associated with each β-lactam using the mutants obtained from the fluctuation assay. Sixty mutants with genetic resistance to a given β-lactam were considered biological replicates and patched onto MIC plates of the two other β-lactams. The patched plates were incubated for 16–20 hr at 37°C. If the mutant grew at MIC of the second β-lactam, it was counted as resistant. If it did not grow at the MIC of the second β-lactam, it was counted as susceptible. For each switch between two drugs, the fraction of cross-resistant mutants was calculated as

$$\frac{Number\ of\ mutants\ that\ grew\ on\ drug\ B}{Total\ mutants\ isolated\ on\ drug\ A}$$

## Statistical analysis for cross-resistance on secondary antibiotic

To test whether the secondary antibiotic had an influence on the degree of cross-resistance of the mutants obtained from the fluctuation assay, we conducted a Fischer's exact test followed by post hoc comparisons using the R package *rcompanion* (*Mangiafico, 2016*). The obtained p-values were then corrected for multiple testing using false discovery rate.

## Statistical analysis of adaptive growth dynamics

To test whether main treatment types were associated with altered dynamics of adaptation in non-extinct populations, we analyzed the trajectories of relative growth yield (as plotted in *Figure 1E* and *Figure 2A*) of drug-treated populations using a GLM, including sequence (##1–16) and transfer as fixed factors and preculture and replicate population as nested random factors (see *Supplementary file 1B* for details). Comparisons between main treatment groups were performed using pairwise post hoc tests and *z* statistics. All p-values were corrected for multiple testing using false discovery rate. The analysis was performed separately for the three time phases 'early' (transfers 2–12), 'middle' (transfers 13–48), and 'late' (transfers 49–96) of the experiment, thus fulfilling the model assumption of response linearity. All statistical analyses were carried out in the statistical environment R (*R Development Core Team, 2020*).

## Statistical analysis of evolved multidrug β-lactam resistance

To test whether evolved populations displayed distinct multidrug β-lactam resistance depending on their main treatment type, we analyzed multidrug β-lactam resistance of evolved isolates – the sum of resistance values against CAR, CEF, and DOR (as plotted in *Figure 3B, C*) – using a GLM. The model included sequence (##1–16) as fixed factor and replicate population as nested random factor (see *Supplementary file 1C* for detailed information). Comparisons between main treatment groups were performed using pairwise post hoc tests and *z* statistics. All p-values were corrected for multiple testing using false discovery rate. The analysis was performed separately for the 'early' (after transfers 12) and 'middle' (transfer 48) time points of the evolution experiment using the R statistical environment (*R Development Core Team, 2020*).

## Statistical analysis of treatment potency predictors

To test whether our experimental (switching rate and temporal irregularity) and biological predictors (hysteresis, probability of direct resistance, and cross effects) were able to explain the variability in our evolutionary responses (extinction, rate of growth adaptation, and multidrug resistance) we carried out a GLM analysis. Values per treatment protocol for the biological predictors were calculated and the GLM analysis then carried out in R (*R Development Core Team, 2020*). We used the *lm* and *anova* commands and the main effects model: response ~ switching rate + irregularity for the experimental predictors and response ~ hysteresis + spontaneous resistance + mutant fraction cross-resistant for the biological predictors.

## Acknowledgements

We would like to thank Leif Tueffers and João Botelho for discussions and suggestions as well as Kira Haas and Julia Bunk for technical support. We acknowledge financial support from the German Science Foundation (grant SCHU 1415/12-2 to HS, and funding under Germany's Excellence Strategy EXC 2167-390884018 as well as the Research Training Group 2501 TransEvo to HS and SN), the Max-Planck Society (IMPRS scholarship to AB; Max-Planck fellowship to HS), and the Leibniz Science Campus Evolutionary Medicine of the Lung (EvoLUNG, to HS and SN). This work was further supported by the German Science Foundation Research Infrastructure NGS_CC (project 407495230) as part of the Next Generation Sequencing Competence Network (project 423957469). NGS analyses were carried out at the Competence Centre for Genomic Analysis Kiel (CCGA Kiel).

## Additional information

### Funding

| Funder | Grant reference number | Author |
|---|---|---|
| Deutsche Forschungsgemeinschaft | SCHU 1415/12-2 | Hinrich Schulenburg |
| Deutsche Forschungsgemeinschaft | EXC 2167-390884018 | Stefan Niemann<br>Hinrich Schulenburg |
| Deutsche Forschungsgemeinschaft | GRK 2501 | Stefan Niemann<br>Hinrich Schulenburg |
| Max-Planck-Gesellschaft | IMPRS Stipend | Aditi Batra |
| Max-Planck-Gesellschaft | Fellowship | Hinrich Schulenburg |
| Leibniz-Gemeinschaft | EvoLUNG | Stefan Niemann<br>Hinrich Schulenburg |
| Deutsche Forschungsgemeinschaft | 407495230 | Sören Franzenburg |

The funders had no role in study design, data collection and interpretation, or the decision to submit the work for publication.

## Author contributions
Aditi Batra, Roderich Roemhild, Conceptualization, Data curation, Formal analysis, Validation, Investigation, Visualization, Methodology, Writing - original draft, Writing - review and editing; Emilie Rousseau, Investigation, Methodology, Writing - review and editing; Sören Franzenburg, Investigation, Project administration, Whole genome sequencing; Stefan Niemann, Supervision, Project administration, Writing - review and editing; Hinrich Schulenburg, Conceptualization, Formal analysis, Supervision, Funding acquisition, Writing - original draft, Project administration, Writing - review and editing

## Author ORCIDs
Roderich Roemhild (iD) http://orcid.org/0000-0001-9480-5261
Stefan Niemann (iD) http://orcid.org/0000-0002-6604-0684
Hinrich Schulenburg (iD) https://orcid.org/0000-0002-1413-913X

## Decision letter and Author response
Decision letter https://doi.org/10.7554/eLife.68876.sa1
Author response https://doi.org/10.7554/eLife.68876.sa2

---

# Additional files

## Supplementary files
• Supplementary file 1. Tables with information on antibiotics used and summaries of the statistical analyses. (**A**) List of antibiotics used for the evolution experiments. (**B**) Statistical analysis of main evolution treatments for the evolutionary dynamics shown in *Figure 2A*. (**C**) Statistical analysis of main evolution treatments for the multidrug β-lactam resistance after transfer 12 in *Figure 2B*. (**D**) Statistical analysis of main evolution treatments for the multidrug β-lactam resistance after transfer 48 in *Figure 2C*. (**E**) Statistical analysis of main evolution treatments for Shannon diversity in *Figure 2B, C*. (**F**) Comparison of resistance profiles between transfer 12 and transfer 48 for *Figure 2—figure supplement 4* using Wilcoxon test and Bonferroni correction[a]. (**G**) Minimum inhibitory concentrations (MICs) as determined by agar dilution for the three β-lactams and as used for the fluctuation assays shown in *Figure 4*. (**H**) Likelihood ratio test to assess pairwise variation in spontaneous resistance rates for the three β-lactam antibiotics shown in *Figure 4B*. (**I**) Post hoc comparisons based on the false discovery rate for phenotype of cross-resistance on secondary antibiotic as shown in *Figure 4C*. (**J**) Analysis of variance of the consequences of the main treatment type on the three measured responses for transfer 12 of the triple β-lactam experiment as summarized in *Figure 5*. (**K**) Post hoc comparison based on the false discovery rate of the effect of the main treatment types on extinction for the triple β-lactam experiment as summarized in *Figure 5*. (**L**) General linear model analysis of the consequences of the experimental predictors switching rate and irregularity on the three measured responses for transfer 12 of the triple β-lactam experiment as summarized in *Figure 5*. (**M**) Main effect tests for the consequences of switching rate and irregularity on extinction and multidrug resistance (MDR) for transfer 12 of the triple β-lactam evolution experiment as summarized in *Figure 5*. (**N**) General linear model analysis of the consequences of the three considered biological predictors, cumulative probability of spontaneous resistance, cross-resistance, and hysteresis on the three measured responses for transfer 12 of the triple β-lactam experiment as summarized in *Figure 5*. (**O**) Main effect tests for the consequences of the three considered biological predictors, cumulative probability of spontaneous resistance, cross-resistance, and hysteresis on extinction of the triple β-lactam evolution experiment as summarized in *Figure 5*.

• Transparent reporting form

## Data availability
Sequencing data have been deposited at NCBI under the BioProject number: PRJNA704789. All other data is provided in the supplementary source data files.

The following dataset was generated:

---

| Author(s) | Year | Dataset title | Dataset URL | Database and Identifier |
|---|---|---|---|---|
| Batra A | 2021 | Sequential beta-lactam treatment genomics | https://www.ncbi.nlm.nih.gov/bioproject/PRJNA704789 | NCBI BioProject, PRJNA704789 |

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
