## [Decision Letter]

**Acceptance summary:**

This study compared effectiveness of three combinations of antibiotic drug therapy regimes against the bacterial pathogen *Pseudomonasaeruginosa* and found that an unlikely drug combination promoted bacterial extinction. In contrast to prior studies using combinations of drugs with similar chemistries (eg. β-lactams that inhibit cell wall synthesis), which lead to cross-resistance and treatment failure, the sequential use of three β-lactams CAR-CEF-DOR (carbenicillin, cefsulodin, doripenem) was highly effective because of constraints caused by DOR. These results revitalize the potential use of similar β-lactams for treatment against bacterial pathogens and promote utility of predicting evolutionary trajectories to improve antimicrobial drug therapies.

**Decision letter after peer review:**

Thank you for submitting your article "High potency of sequential therapy with only beta-lactam antibiotics" for consideration by eLife. Your article has been reviewed by 3 peer reviewers, including Vaughn S Cooper as the Reviewing Editor and Reviewer #1, and the evaluation has been overseen by George Perry as the Senior Editor.

Essential Revisions:

1) The sole strain used in this work is *P. aeruginosa* PA14. This is a good strain choice but the paper seems to presume that this isolate represents the species well, or even more broadly. What is known in the literature about variation within PA or related species in responses to DOR, and/or other drugs in the triple combinations? Is the hysteresis effect limited to this strain?

2) Given that much of the paper focuses on the doripenem effect, what are the relevant attributes of this drug relative to others? At face value, it's a long-lived carbapenem, so is it the fact that it's a carbapenem, its durability, or both the cause of this physiological stress that diminishes subsequent survival? Might the hysteresis effect simply be the killing of susceptible cells causing subsequent failure?

3) A primary conclusion of this paper (L 359-61) is that the spontaneous rate of resistance was a key variable dictating treatment potency. However, this conclusion does not appear to account for the diminished population size of DOR-treated populations and the high potency of this drug relative to others, which would increase the likelihood of error in DOR dosage. While Figure 4 shows that mutation rates to DOR are low, and the methods are appropriate, the results could reflect stronger selection imposed by DOR concentrations and smaller genomic targets to resistance. The authors use this to make their case, but do not adequately discuss the differences between these three β-lactam drugs. In other words, I wonder if PA14 is on a knife-edge in the presence of DOR at the concentrations used, and this reduces population sizes, strengthens selection beyond other drugs, and leads to more variable extinction probabilities. Discussion of the population-genetic consequences of evolution at 0.75x the MIC, and the role of error around this 0.75x threshold, are needed.

4) Figure 1A: I found the different dashed lines to be difficult to distinguish. Maybe place each label near each closed contour so it's easy to see? Also, consider emphasizing those from this study (perhaps with thicker contours?). The positioning of the drug classes makes it unclear what STR is. Is it DNA gyrase or Ribosome since it is in the middle of those terms? Perhaps use color coding. Also, please define what FQ, AG and BL are in Figure 1A.

5) Lines 47-50: It is unclear how competitive release occurs in this scenario. It is my understanding that competitive release occurs when one lineage acquires a highly beneficial mutation early that leads to competitive exclusion of the others. And the paper cited suggests that this phenomenon occurs due to selective pressure not strong enough to prevent this mutation. However, how it is written suggests that easily accessible genetic resistance is the source. But if it were indeed easily accessible, then there would be higher chances that other lineages would also acquire it? If my understanding is incorrect, perhaps provide a bit more of an explanation for the term competitive release.

6) Line 80-84: It is unclear how negative hysteresis was suppressed by "strong selection dynamics" with a mild increase in resistance. Can you provide a specific genetic example from the paper? This will also help conceptualize types of negative hysteresis that can occur during strong antibiotic pressures.

7) Line 91: Replace "As before" with "The new" and add "expected to contribute to collateral sensitivity…". As written, it is confusing what was previously tested and what the new design and expectations are in this study.

8) Line 97: Similar to above. Add "expected to be dominated by cross-resistance…" Helps to clarify past from present.

9) The term "season" is confusing and often used interchangeably with "transfer". I prefer the term transfer and suggest keeping this terminology throughout.

10) Figure Supplemental 1, Table1: I like the extra information about the stock concentration and storage. It would also be helpful if the drugs were ordered by class and maybe even identified using the BL, FQ, AG abbreviations.

11) Figure 2: Please describe that the dotted lines represent the three growth phases in A. For B, it is unclear what the arrows signify. It looks like they are aligned with transfers 12 and 48. Either make them more continuous or remove them altogether. Otherwise, it looks as if the populations are what is being highlighted. Also, what is the significance of the +1 to -0.5 range vs the 16x to 0.33x? The MIC column is most explanatory to me.

12) Line 192-193: To my eye, the resistance profiles between the early- and mid- phases look pretty similar. The sum of resistances is mentioned. Perhaps add the sum to each of the columns in Figure 2 to make it clearer.

13) Line 210-213: DOR monotherapy is stated to not show a noticeable amount of resistance at the early stage. However, it looks like population 1 shows 4x-8X MIC and looks similar to the mid phase. This is contradictory to what is stated. Can you please rephrase.

14) Line 210-213: It is interesting that monotherapy population one at transfer 12 showed increased resistance but resulted in no variants found. Is it possible that this was a result of heteroresistance or persistence? If you sequenced the whole population instead of isolating three clones, you might be able to distinguish variants at low frequency that would contribute to heteroresistance.

15) Figure 3C. The Blue and black are hard to distinguish as a gradient.

16) Line 361: Is it treatment potency due to cross resistance or the lack of cross-resistance. I thought that it was due to the latter because neither of the other two drugs promoted cross-resistance to DOR.

17) Line 414: Is "disadvantageous cross-resistance" the same as collateral sensitivity? If so, the double negative is confusing.

Recommendations for follow up work

1. Whole genome sequencing was limited to three clones per population. While it is understandable that the number of treatment regimens might make sequencing costly. Population dynamics and allele frequencies may help explain why the population during the DOR monotherapy struggled to acquire mutations before transfer 12.

2. One of the most interesting aspects of the study is that the principles identified for designing sequential therapies are not dependent on a unique molecular mechanism, and one could therefore envision multiple drug combinations that achieve similar evolution-slowing effects but do so in very different ways at the molecular level (where resistance has often been studied). Do the authors believe there is some inherent feature of β-lactams-for example, the fact that they compromise the cell wall and promote lysis-that may underlie the success of homogeneous sequential therapies? Or might one expect similar success with, say, triplets of fluoroquinolones or other drugs, as long as the triplet is chosen with similar qualitative features (e.g. low levels of cross-resistance, low spontaneous resistance to at least one drug). Relatedly, are there reasons to believe that certain classes of drugs will be more likely to show those qualitative features? It would be remarkable if, for example, these important features could be predicted based on, say, the known diversity of resistant mutations for the component drugs. In any event, I think one of the striking conclusions from this work is that considerable optimization may be possible without detailed knowledge of molecular mechanism.

3. The results suggest that the chosen β lactams exhibit negative hysteresis, yet the negative hysteresis does not play a dominant role in slowing resistance (unlike in the authors' previous study with heterogeneous drug combinations). Does this discrepancy arise simply because other effects (e.g. collateral effects) are dominant, so the impacts of negative hysteresis are (by comparison) negligible? Or is there some feature of the negative hysteresis-and perhaps its correlation with the collateral effects-that renders it non-beneficial? I'd venture a guess that in general, there is some underlying relationship between the hysteresis profiles, the collateral profiles, and the timescale of switching that could conceivably be tuned to optimize these therapies. It seems to be an exciting topic for future work.

*Reviewer #1 (Recommendations for the authors):*

The authors conducted an ambitious set of hundreds of evolution experiments with *P. aeruginosa* exposed to different sets of 3 antibiotics amounting to ~500 generations each, under monotherapy, fast switching, slow switching, and random switching. A primary outcome was extinction, ie failure to persist, while secondary outcomes included the extent and breadth of evolved resistance and a sampling of evolved resistance mutations. Treatments including the antibiotic doripenem led to more extinctions, likely because growth in doripenem reduced the ability to grow in alternative antibiotics, which they term negative hysteresis. This is an interesting and potentially clinically relevant discovery. The paper is well written and engaging.

I see three major limitations that the authors could reasonably address.

1. The sole strain used in this work is *P. aeruginosa* PA14. This is a good strain choice but the paper seems to presume that this isolate represents the species well, or even more broadly. What is known in the literature about variation within PA or related species in responses to DOR, and/or other drugs in the triple combinations? Is the hysteresis effect limited to this strain?

2. Given that much of the paper focuses on the doripenem effect, what are the relevant attributes of this drug relative to others? At face value, it's a long-lived carbapenem, so is it the fact that it's a carbapenem, its durability, or both the cause of this physiological stress that diminishes subsequent survival? Might the hysteresis effect simply be the killing of susceptible cells causing subsequent failure?

3. A primary conclusion of this paper (L 359-61) is that the spontaneous rate of resistance was a key variable dictating treatment potency. However, this conclusion does not appear to account for the diminished population size of DOR-treated populations and the high potency of this drug relative to others, which would increase the likelihood of error in DOR dosage. While Figure 4 shows that mutation rates to DOR are low, and the methods are appropriate, the results could reflect stronger selection imposed by DOR concentrations and smaller genomic targets to resistance. The authors use this to make their case, but do not adequately discuss the differences between these three β-lactam drugs. In other words, I wonder if PA14 is on a knife-edge in the presence of DOR at the concentrations used, and this reduces population sizes, strengthens selection beyond other drugs, and leads to more variable extinction probabilities. Discussion of the population-genetic consequences of evolution at 0.75x the MIC, and the role of error around this 0.75x threshold, are needed.

*Reviewer #2 (Recommendations for the authors):*

Batra, Aditi et al., compared the effectiveness of three in vitro antibiotic drug therapy regimes against the bacterial pathogen *Pseudomonasaeruginosa* and found that an unlikely drug combination promoted bacterial extinction. Previous studies have shown that using similar drug classes with similar chemistries (eg. β-lactams that inhibit cell wall synthesis) lead to increased cross resistance and therefore, treatment failure. However, the authors found that the fast and sequential use of three β-lactams CAR-CEF-DOR (carbenicillin, cefsulodin, doripenem) was the most effective treatment against *P. aeruginosa*. The authors further investigated the potential factors contributing to a lack of evolutionary rescue such as negative hysteresis, collateral sensitivity, and rate of spontaneous direct/indirect resistance. By conducting additional experiments to test multiple hypotheses, the authors concluded that the fast-switching three-drug series was constrained by DOR and that the cause of the constraint was due to a low rate of spontaneous resistance and cross-resistance to DOR. These results revitalize the potential use of similar β-lactams for treatment against bacterial pathogens and the utility of predicting evolutionary trajectories to improve antimicrobial drug therapies.

Strengths:

This paper introduces both physiological and evolutionary concepts as possible hypotheses for explaining antibiotic drug constraints. When discussing consequences to drug therapies, cross resistance and collateral sensitivity are often described whereas physiological constraints such as negative hysteresis is overlooked. The experiments conducted in this paper demonstrate how to test for hysteresis using short exposures to a drug and then switching to a different drug and comparing growth killing effects. Drug exposure is carried out quickly to maintain a low mutational load and reduce the probability that a beneficial mutation would arise. Mutations would interfere with a test for non-genetic physiological conditioning.

Measurements of spontaneous resistance rates is a useful phenotype to compare mechanisms of mutations for each drug. The experiments and results that compare the rate of spontaneous direct resistance and indirect (ie. Cross resistance) as well as collateral sensitivity are clear and well replicated.

Weakness:

Whole genome sequencing was limited to three clones per population. While it is understandable that the number of treatment regimens might make sequencing costly. Population dynamics and allele frequencies may help explain why the population during the DOR monotherapy struggled to acquire mutations before transfer 12.*Reviewer #3 (Recommendations for the authors):*

The manuscript by Batra and colleagues investigates the impact of sequential antibiotic treatments on evolving laboratory populations of *P. aeruginosa*, a clinically important microbial pathogen. A growing body of work suggests that cycling between different antibiotics can slow resistance in the lab, though the conventional view is that successful treatments would require drugs from different classes, limiting the propensity for cross resistance and potentially harnessing sensitizing collateral effects or negative hysteresis. This study provides a large-scale investigation of the efficacy of sequential treatments comprised of different 3-drug treatments. The drugs are chosen to span different classes, with triplicates including drugs targeting the ribosome, DNA gyrase, or the cell wall (β lactams) in some combination. Previous work from the same group showed that fast switching between a heterogeneous set (3 different classes) of drugs promoted population extinction at sub-MIC concentrations, with selection favoring genetic suppression of negative hysteresis. In this work, they investigate the response to different 3-drug sequences (both homogeneous and heterogeneous) in a total of 756 (!!) replicate populations. Treatments included monotherapy, fast or slow switching, and random switching. Interestingly, they found that switching among two different sets of β lactams produced high levels of extinction-comparable to that previously observed with a heterogeneous set-while switching between a new heterogeneous set did not produce extinction. Extinction tended to occur early in the treatment and was significantly reduced when the period of treatment switching was extended. The authors focus on one drug set (CAR-CEF-DOR) for an in-depth characterization. Using general linear models (GLM) to evaluate growth adaptation and total β lactam resistance in the three distinct growth phases, they showed that early phase adaptation is decelerated in the fast-switching group relative to monotherapy and slow-switching group; interestingly, the fast and random treatments showed relatively similar rates of extinction, suggesting that precise ordering is not the primary driver of the effects. They also observed generally increased β lactam resistance in the second phase, but the different treatment types led to largely distinct resistance profiles. Population diversity increased between early (transfer 12) and later (transfer 48) days of adaptation, and resistance in the early phase was often constrained by susceptibility to a particular drug (DOR) (and whole-genome sequencing found DOR resistance mutations only at later times). The authors go on to show that negative hysteresis is common among the three β-lactams (Figure 3), while DOR exhibits the lowest rates of direct and cross resistance (as measured by classic fluctuation tests and a patching assay). Finally, they used another GLM to quantify the impacts of different features on slowing resistance; they found that switching rate, temporal irregularity, spontaneous resistance probability, and (most prominently) the collateral effects dominate, with low levels of spontaneous resistance to DOR a primary constraint.

This study is timely, important, extremely thorough, and perhaps most importantly, it helps to identify a number of new principles to guide the rational design of sequential antibiotic therapies. It represents a monumental effort to thoroughly characterize adaption to 3-drug cycles in several representative examples, an achievement in itself. But as impressively, the authors draw on GLM's to extract meaningful patterns from the web of complex phenotypic data, and in doing so they uncover features of these treatments (e.g. surprising cross-resistance landscapes between drugs from the same class; potential importance of having one drug with low rates of spontaneous resistance) that may guide general design of new therapies. As with all good studies, it raises a number of questions for future work, and I list a few that come to mind below (that authors may wish to respond, perhaps in the discussion, at their discretion). But in general, I have no major concerns with the study at a technical or conceptual level (a rarity!). I believe this is an excellent contribution to the literature that will interest a broad range of readers in evolutionary biology and microbiology.

1) To me, one of the most interesting aspects of the study is that the principles identified for designing sequential therapies are not dependent on a unique molecular mechanism, and one could therefore envision multiple drug combinations that achieve similar evolution-slowing effects but do so in very different ways at the molecular level (where resistance has often been studied). Do the authors believe there is some inherent feature of β-lactams-for example, the fact that they compromise the cell wall and promote lysis-that may underlie the success of homogeneous sequential therapies? Or might one expect similar success with, say, triplets of fluoroquinolones or other drugs, as long as the triplet is chosen with similar qualitative features (e.g. low levels of cross-resistance, low spontaneous resistance to at least one drug). Relatedly, are there reasons to believe that certain classes of drugs will be more likely to show those qualitative features? It would be remarkable if, for example, these important features could be predicted based on, say, the known diversity of resistant mutations for the component drugs. In any event, I think one of the striking conclusions from this work is that considerable optimization may be possible without detailed knowledge of molecular mechanism.

2) The results suggest that the chosen β lactams exhibit negative hysteresis, yet the negative hysteresis does not play a dominant role in slowing resistance (unlike in the authors' previous study with heterogeneous drug combinations). Does this discrepancy arise simply because other effects (e.g. collateral effects) are dominant, so the impacts of negative hysteresis are (by comparison) negligible? Or is there some feature of the negative hysteresis-and perhaps its correlation with the collateral effects-that renders it non-beneficial? I'd venture a guess that in general, there is some underlying relationship between the hysteresis profiles, the collateral profiles, and the timescale of switching that could conceivably be tuned to optimize these therapies. It seems to be an exciting topic for future work.

---

## [Author Response]

Essential Revisions:1) The sole strain used in this work is *P. aeruginosa* PA14. This is a good strain choice but the paper seems to presume that this isolate represents the species well, or even more broadly. What is known in the literature about variation within PA or related species in responses to DOR, and/or other drugs in the triple combinations? Is the hysteresis effect limited to this strain?

Many thanks for this comment. We agree that the strain PA14 does not necessarily capture the phenotypes present within the species *Pseudomonas aeruginosa* and it is also possible that other bacterial species may show different evolutionary responses to the treatments tested. We here use PA14 as an established, tractable pathogen model system. In response to the reviewer’s comment, we now emphasize this point at the end of the introduction. Moreover, some information is indeed available on the response of other *P. aeruginosa* strains to the three antibiotics used. We now summarized this information in a new paragraph in the discussion. In contrast, hysteresis tests with the considered three antibiotics have thus far only been done by us, and the results are presented in the manuscript. It would indeed be of particular value to assess such hysteresis also in other strains, as we now point out in the final part of the discussion. In fact, this is something which we will address in a new project in my lab.

2) Given that much of the paper focuses on the doripenem effect, what are the relevant attributes of this drug relative to others? At face value, it's a long-lived carbapenem, so is it the fact that it's a carbapenem, its durability, or both the cause of this physiological stress that diminishes subsequent survival? Might the hysteresis effect simply be the killing of susceptible cells causing subsequent failure?

This is indeed a very interesting point. We now added more information on the characteristics of Doripenem in comparison with the other drugs in the first part of the discussion. Moreover, as indicated above, we now also provide a focused summary of the available information on resistance rate evolution of *P. aeruginosa* to Doripenem and the other two drugs. As to hysteresis: The used assay is set up to minimize any killing or differential replication of cells during the pretreatment, in order to ensure that any responses seen in the main treatment are based on physiological changes and not selection. This aspect is now highlighted in the Results section in the chapter on hysteresis.

3) A primary conclusion of this paper (L 359-61) is that the spontaneous rate of resistance was a key variable dictating treatment potency. However, this conclusion does not appear to account for the diminished population size of DOR-treated populations and the high potency of this drug relative to others, which would increase the likelihood of error in DOR dosage. While Figure 4 shows that mutation rates to DOR are low, and the methods are appropriate, the results could reflect stronger selection imposed by DOR concentrations and smaller genomic targets to resistance. The authors use this to make their case, but do not adequately discuss the differences between these three β-lactam drugs. In other words, I wonder if PA14 is on a knife-edge in the presence of DOR at the concentrations used, and this reduces population sizes, strengthens selection beyond other drugs, and leads to more variable extinction probabilities. Discussion of the population-genetic consequences of evolution at 0.75x the MIC, and the role of error around this 0.75x threshold, are needed.

Many thanks for this interesting point. However, we politely disagree that a stronger reduction in population size through DOR in comparison to the other drugs could have influenced the results.

Firstly, the dose response curves in the first supplementary figure (Figure 1—figure supplement 1) indicate very little variation in yield upon DOR treatment, especially at the IC75 concentration used to initiate the evolution experiments. Please note the 2nd panel in the top row and the right panel in the middle row; error bars are standard deviation of the mean (using 6 biological replicates) and hardly visible for the IC75 point because of the lack of variation (see red data points). Much more variation was for example found for CAR.

Secondly, the main evolution experiments were initiated using specifically standardized IC75 drug concentrations. Therefore, we do not expect initial variation in population size across treatments. Such variation should only arise thereafter, as a consequence of hysteresis effects or spontaneously evolved resistances and collateral effects.

Thirdly, the same argument holds for the fluctuation assays, which were performed with specifically standardized antibiotic concentrations, leading to identical selection being imposed to capture the spontaneous mutants.

Therefore, we consider it most likely that the lower adaptation to DOR in the monotherapies and the DOR transfer periods within the sequential treatments and also the reduced resistance rate within the fluctuation assays are due to a lower rate of DOR-specific resistance mutations (i.e., smaller genomic targets to DOR resistance) and also a lower rate of cross-resistance to DOR. These arguments are now presented in the revised Results section – in the chapter on the dynamics of resistance for the triple β-lactam treatment, and the chapter on spontaneous resistance rates.

4) Figure 1A: I found the different dashed lines to be difficult to distinguish. Maybe place each label near each closed contour so it's easy to see? Also, consider emphasizing those from this study (perhaps with thicker contours?). The positioning of the drug classes makes it unclear what STR is. Is it DNA gyrase or Ribosome since it is in the middle of those terms? Perhaps use color coding. Also, please define what FQ, AG and BL are in Figure 1A.

Many thanks for your suggestions. We hope that the revised figure is more accessible.

5) Lines 47-50: It is unclear how competitive release occurs in this scenario. It is my understanding that competitive release occurs when one lineage acquires a highly beneficial mutation early that leads to competitive exclusion of the others. And the paper cited suggests that this phenomenon occurs due to selective pressure not strong enough to prevent this mutation. However, how it is written suggests that easily accessible genetic resistance is the source. But if it were indeed easily accessible, then there would be higher chances that other lineages would also acquire it? If my understanding is incorrect, perhaps provide a bit more of an explanation for the term competitive release.

We agree that the sentence was not entirely clear. We have now removed the concept of competitive release from the sentence, as it is not important for the particular argument given and also not for the focus of our paper.

6) Line 80-84: It is unclear how negative hysteresis was suppressed by "strong selection dynamics" with a mild increase in resistance. Can you provide a specific genetic example from the paper? This will also help conceptualize types of negative hysteresis that can occur during strong antibiotic pressures.

Many thanks for helping us to clarify this section. We now refer to a genetic example from our previous publication in the revised text.

7) Line 91: Replace "As before" with "The new" and add "expected to contribute to collateral sensitivity…". As written, it is confusing what was previously tested and what the new design and expectations are in this study.

Many thanks. This is now corrected.

8) Line 97: Similar to above. Add "expected to be dominated by cross-resistance…" Helps to clarify past from present.

Corrected.

9) The term "season" is confusing and often used interchangeably with "transfer". I prefer the term transfer and suggest keeping this terminology throughout.

Many thanks for this suggestion, which we now followed throughout the revised manuscript.

10) Figure Supplemental 1, Table1: I like the extra information about the stock concentration and storage. It would also be helpful if the drugs were ordered by class and maybe even identified using the BL, FQ, AG abbreviations.

We followed the advice of the reviewer and changed the table accordingly.

11) Figure 2: Please describe that the dotted lines represent the three growth phases in A. For B, it is unclear what the arrows signify. It looks like they are aligned with transfers 12 and 48. Either make them more continuous or remove them altogether. Otherwise, it looks as if the populations are what is being highlighted. Also, what is the significance of the +1 to -0.5 range vs the 16x to 0.33x? The MIC column is most explanatory to me.

Many thanks for your suggestions.

First, we now describe that the dotted lines indicate the three growth phases in the text (line 165-166) and in the figure legend.

Second, we have replaced the misleading arrows by curled brackets to more clearly indicate that all adjacent columns characterize the same transfer (see revised Figure 2).

Third, we completely agree with the reviewer that the MIC column of the resistance scale is more intuitive. However, the visualized resistance scale that ranges from -0.5 to +1 is a better representation of the observed biological variation as it captures not only the x-axis intercept, but the entire shape of the dose-response curve. Thus, the used parameter will differentiate sub-populations that have equivalent MIC, but differ in dose-sensitivity or metabolic fitness cost. Still, it is important to know the approximate range of effect sizes, for which we indicated approximate change of MIC corresponding to resistance value. In the revised manuscript, we now provide more information about the double scaling of the resistance quantification in the new Figure 2—figure supplement 3.

12) Line 192-193: To my eye, the resistance profiles between the early- and mid- phases look pretty similar. The sum of resistances is mentioned. Perhaps add the sum to each of the columns in Figure 2 to make it clearer.

Many thanks for this comment. The reviewer is correct that there is no general increase in resistance across the two considered time points. Instead, resistance changes vary and depend on the exact treatment. In response to the reviewer’s comments, we now adjusted our description of the results, where we now point out that “Resistance to the used β-lactams increased across the two time points only in some treatments, but not all (Figure 2B and C, Figure 2—figure supplement 4, Figure 2—figure supplement 5, Supplementary File 1F), suggesting treatment-dependent evolutionary responses to the antibiotics.”. We also adjusted our conclusions of this part of the result, where we now state that “the population analysis of resistance profiles indicates that resistance evolution depends on the exact treatment protocol and that the dynamics of resistance emergence to DOR may be key for the observed deceleration of beta-lactam adaptation in the fast-regular treatments.”. To enhance clarity, we further added two supplementary figures.

Figure 2—figure supplement 4 shows the early- and mid-phase resistance values side-by-side for each antibiotic in a box plot design. Please see also the related new statistical analysis in Supplementary File 1F that shows which population resistance profiles changed over time.

The sum of resistances is now presented in Figure 2—figure supplement 5, separately for the two considered transfer time points in panels A and B and also separately for three β-lactams (left columns) and the two collateral antibiotics CIP and GEN (right columns). We now refer to this figure to further support our observation of the comparatively high efficacy of irregular sequential protocols in the discussion.

13) Line 210-213: DOR monotherapy is stated to not show a noticeable amount of resistance at the early stage. However, it looks like population 1 shows 4x-8X MIC and looks similar to the mid phase. This is contradictory to what is stated. Can you please rephrase.

Many thanks for this comment. It is based on a misunderstanding of the design of panels B and C of Figure 2, which were apparently not sufficiently clear. In detail, Mono 1 is monotherapy with CAR, mono 2 is monotherapy with DOR, and mono 3 is monotherapy with CEF. For the early time point, the considered isolates from DOR treatment (=2nd row) did not produce any resistances against DOR (=2nd column) and almost no resistance against any of the other drugs. In the revised manuscript, we now describe the design of the figure in more detail in order to prevent any misunderstandings.

14) Line 210-213: It is interesting that monotherapy population one at transfer 12 showed increased resistance but resulted in no variants found. Is it possible that this was a result of heteroresistance or persistence? If you sequenced the whole population instead of isolating three clones, you might be able to distinguish variants at low frequency that would contribute to heteroresistance.

Many thanks for this comment. It shows that our figure was not sufficiently clear, as the comment is based on the same misunderstanding as mentioned above for comment 13. In detail, the results for the DOR monotherapy are shown in the second row of panels B and C of Figure 3 (not the first row). Thus, the isolates from the DOR treatment at the early time point do not show increases in resistance. This is then entirely consistent with our finding that the sequenced genomes for this treatment at the early time point do not bear any resistance mutations. This is something, which we now emphasize in the revised manuscript. These consistent results indicate that evolution of resistance to DOR is constrained, possibly because of a smaller range of genomic targets for DOR resistance. We apologize that the design of Figure 2 panels B and C led to a misunderstanding. As outlined above, we adjusted the design and description of the two panels, in order to prevent such misunderstandings.

We further agree with the reviewer that a more detailed population genomic analysis could have provided further insights into adaptation to DOR monotherapy. The conducted sequencing does not have the resolution to detect low-frequency variants or potential gene amplifications that could confer heteroresistance. However, we are also convinced that such additional genomics data is not essential for our conclusions. Firstly, the phenotypic and genomic results are consistent, as outlined above. Secondly, the indicated lower mutation rate towards DOR resistance was independently validated by fluctuation assays. And this test did indeed confirm the initial finding. This is a point, which we now emphasize more clearly in the Results section. Therefore, we are convinced that our conclusion of a constrained rate of DOR resistance evolution is fully valid and justified and would not receive further support by population sequencing.

Nevertheless, we agree that an understanding of the very early steps of antibiotic adaptation is very important. We are therefore considering to perform a detailed population genomic sequence analysis in the future. Unfortunately, we are currently constrained by consequences of the Corona pandemic, because our sequencing center cannot promise results in the close future. Hence, we are grateful that the *eLife* guidelines suggest to perform such interesting further analyses separate from the current manuscript with the possibility of linking them in the future. Most importantly, as outlined above, the suggested population genome analyses would be a useful addition, but they are not essential for the final conclusions drawn.

15) Figure 3C. The Blue and black are hard to distinguish as a gradient.

The color scheme has been adjusted to improve clarity.

16) Line 361: Is it treatment potency due to cross resistance or the lack of cross-resistance. I thought that it was due to the latter because neither of the other two drugs promoted cross-resistance to DOR.

Many thanks for this comment. We agree that our previous statement was not sufficiently clear. We now point out that treatment potency was determined by variation in the spontaneous rate of resistance to the β-lactams and by variation in the resulting collateral effects across sequential treatment protocols.

17) Line 414: Is "disadvantageous cross-resistance" the same as collateral sensitivity? If so, the double negative is confusing.

This expression is indeed not very clear. We deleted the word “disadvantageous”, because it is not really needed here.

Recommendations for follow up work1. Whole genome sequencing was limited to three clones per population. While it is understandable that the number of treatment regimens might make sequencing costly. Population dynamics and allele frequencies may help explain why the population during the DOR monotherapy struggled to acquire mutations before transfer 12.

Many thanks for this recommendation. We agree that a more detailed population genomic analysis of the evolving bacteria could yield important insights into the exact dynamics of resistance evolution. We are indeed planning to perform such analyses in the future – as soon as this is reliably possible again, and we would then be happy to link the results to the current manuscript. We would still like to point out that these population genomic analyses will be insightful, yet they are not essential for our conclusion on the constrained rate of DOR resistance evolution, as explained in detail in our above reply.

2. One of the most interesting aspects of the study is that the principles identified for designing sequential therapies are not dependent on a unique molecular mechanism, and one could therefore envision multiple drug combinations that achieve similar evolution-slowing effects but do so in very different ways at the molecular level (where resistance has often been studied). Do the authors believe there is some inherent feature of β-lactams-for example, the fact that they compromise the cell wall and promote lysis-that may underlie the success of homogeneous sequential therapies? Or might one expect similar success with, say, triplets of fluoroquinolones or other drugs, as long as the triplet is chosen with similar qualitative features (e.g. low levels of cross-resistance, low spontaneous resistance to at least one drug). Relatedly, are there reasons to believe that certain classes of drugs will be more likely to show those qualitative features? It would be remarkable if, for example, these important features could be predicted based on, say, the known diversity of resistant mutations for the component drugs. In any event, I think one of the striking conclusions from this work is that considerable optimization may be possible without detailed knowledge of molecular mechanism.

Many thanks for these comments. We agree that the raised questions would be extremely fascinating to study in more detail. They cannot really be answered based on the current data. In fact, we are in the process of setting up new projects that address for example the last point and also the molecular basis of interactions between different β-lactam drugs. We will hopefully be able to present interesting results on at least these points in the future.

3. The results suggest that the chosen β lactams exhibit negative hysteresis, yet the negative hysteresis does not play a dominant role in slowing resistance (unlike in the authors' previous study with heterogeneous drug combinations). Does this discrepancy arise simply because other effects (e.g. collateral effects) are dominant, so the impacts of negative hysteresis are (by comparison) negligible? Or is there some feature of the negative hysteresis-and perhaps its correlation with the collateral effects-that renders it non-beneficial? I'd venture a guess that in general, there is some underlying relationship between the hysteresis profiles, the collateral profiles, and the timescale of switching that could conceivably be tuned to optimize these therapies. It seems to be an exciting topic for future work.

Again, very interesting points. And again, we cannot answer the raised questions in consideration of the current data. There is clearly a need for an even more systematic analysis on the relationship between hysteresis, collateral effects, and the rate of antibiotic changes. As part of the above mentioned new projects, we do indeed plan to study the molecular basis of hysteresis so that this information could be linked to known mechanisms of collateral sensitivity. So again, we hope to be able to answer at least some of the raised questions with our new projects in the future.

Reviewer #1 (Recommendations for the authors):The authors conducted an ambitious set of hundreds of evolution experiments with *P. aeruginosa* exposed to different sets of 3 antibiotics amounting to ~500 generations each, under monotherapy, fast switching, slow switching, and random switching. A primary outcome was extinction, ie failure to persist, while secondary outcomes included the extent and breadth of evolved resistance and a sampling of evolved resistance mutations. Treatments including the antibiotic doripenem led to more extinctions, likely because growth in doripenem reduced the ability to grow in alternative antibiotics, which they term negative hysteresis. This is an interesting and potentially clinically relevant discovery. The paper is well written and engaging.I see three major limitations that the authors could reasonably address.1. The sole strain used in this work is *P. aeruginosa* PA14. This is a good strain choice but the paper seems to presume that this isolate represents the species well, or even more broadly. What is known in the literature about variation within PA or related species in responses to DOR, and/or other drugs in the triple combinations? Is the hysteresis effect limited to this strain?

Many thanks for this comment. We agree that the strain PA14 does not necessarily capture the phenotypes present within the species *Pseudomonas aeruginosa* and it is also possible that other bacterial species may show different evolutionary responses to the treatments tested. We here use PA14 as a tractable model system. In response to the reviewer’s comment, we now emphasize this point at the end of the introduction. Moreover, some information is indeed available on the response of other *P. aeruginosa* strains to the three antibiotics used. We now summarized this information in a new paragraph in the discussion. In contrast, hysteresis tests with the considered three antibiotics have thus far only been done by us, and the results are presented in the manuscript. It would indeed be of particular value to assess such hysteresis also in other strains, as we now point out in the final part of the discussion.

2. Given that much of the paper focuses on the doripenem effect, what are the relevant attributes of this drug relative to others? At face value, it's a long-lived carbapenem, so is it the fact that it's a carbapenem, its durability, or both the cause of this physiological stress that diminishes subsequent survival? Might the hysteresis effect simply be the killing of susceptible cells causing subsequent failure?

This is indeed a very interesting point. We now added more information on the characteristics of Doripenem in comparison with the other drugs in the first part of the discussion. Moreover, as indicated above, we now also provide a focused summary of the available information on resistance rate evolution of *P. aeruginosa* to Doripenem and the other two drugs. As to hysteresis: The used assay is set up to minimize any killing or differential replication of cells during the pretreatment, in order to ensure that any responses seen in the main treatment are based on physiological changes and not selection. This aspect is now highlighted in the Results section in the chapter on hysteresis.

3. A primary conclusion of this paper (L 359-61) is that the spontaneous rate of resistance was a key variable dictating treatment potency. However, this conclusion does not appear to account for the diminished population size of DOR-treated populations and the high potency of this drug relative to others, which would increase the likelihood of error in DOR dosage. While Figure 4 shows that mutation rates to DOR are low, and the methods are appropriate, the results could reflect stronger selection imposed by DOR concentrations and smaller genomic targets to resistance. The authors use this to make their case, but do not adequately discuss the differences between these three β-lactam drugs. In other words, I wonder if PA14 is on a knife-edge in the presence of DOR at the concentrations used, and this reduces population sizes, strengthens selection beyond other drugs, and leads to more variable extinction probabilities. Discussion of the population-genetic consequences of evolution at 0.75x the MIC, and the role of error around this 0.75x threshold, are needed.

Many thanks for this interesting point. However, we politely disagree that a stronger reduction in population size through DOR in comparison to the other drugs could have influenced the results.

Firstly, the dose response curves in the first supplementary figure (Figure 1—figure supplement 1) indicate very little variation in yield upon DOR treatment, especially at the IC75 concentration used to initiate the evolution experiments. Please note the 2nd panel in the top row and the right panel in the middle row; error bars are standard deviation of the mean (using 6 biological replicates) and hardly visible for the IC75 point because of the lack of variation (see red data points). Much more variation was for example found for CAR.

Secondly, the main evolution experiments were initiated using specifically standardized IC75 drug concentrations. Therefore, we do not expect initial variation in population size across treatments. Such variation should only arise thereafter, as a consequence of hysteresis effects or spontaneously evolved resistances and collateral effects.

Thirdly, the same argument holds for the fluctuation assays, which were performed with specifically standardized antibiotic concentrations, leading to identical selection being imposed to capture the spontaneous mutants.

Therefore, we consider it most likely that the lower adaptation to DOR in the monotherapies and the DOR transfer periods within the sequential treatments and also the reduced resistance rate within the fluctuation assays are due to a lower rate of DOR-specific resistance mutations (i.e., smaller genomic targets to DOR resistance) and also a lower rate of cross-resistance to DOR. These arguments are now presented in the revised Results section – in the chapter on the dynamics of resistance for the triple β-lactam treatment, and the chapter on spontaneous resistance rates.

Reviewer #2 (Recommendations for the authors):Batra, Aditi et al., compared the effectiveness of three in vitro antibiotic drug therapy regimes against the bacterial pathogen *Pseudomonas aeruginosa* and found that an unlikely drug combination promoted bacterial extinction. Previous studies have shown that using similar drug classes with similar chemistries (eg. β-lactams that inhibit cell wall synthesis) lead to increased cross resistance and therefore, treatment failure. However, the authors found that the fast and sequential use of three β-lactams CAR-CEF-DOR (carbenicillin, cefsulodin, doripenem) was the most effective treatment against *P. aeruginosa*. The authors further investigated the potential factors contributing to a lack of evolutionary rescue such as negative hysteresis, collateral sensitivity, and rate of spontaneous direct/indirect resistance. By conducting additional experiments to test multiple hypotheses, the authors concluded that the fast-switching three-drug series was constrained by DOR and that the cause of the constraint was due to a low rate of spontaneous resistance and cross-resistance to DOR. These results revitalize the potential use of similar β-lactams for treatment against bacterial pathogens and the utility of predicting evolutionary trajectories to improve antimicrobial drug therapies.Strengths:This paper introduces both physiological and evolutionary concepts as possible hypotheses for explaining antibiotic drug constraints. When discussing consequences to drug therapies, cross resistance and collateral sensitivity are often described whereas physiological constraints such as negative hysteresis is overlooked. The experiments conducted in this paper demonstrate how to test for hysteresis using short exposures to a drug and then switching to a different drug and comparing growth killing effects. Drug exposure is carried out quickly to maintain a low mutational load and reduce the probability that a beneficial mutation would arise. Mutations would interfere with a test for non-genetic physiological conditioning.Measurements of spontaneous resistance rates is a useful phenotype to compare mechanisms of mutations for each drug. The experiments and results that compare the rate of spontaneous direct resistance and indirect (ie. Cross resistance) as well as collateral sensitivity are clear and well replicated.Weakness:Whole genome sequencing was limited to three clones per population. While it is understandable that the number of treatment regimens might make sequencing costly. Population dynamics and allele frequencies may help explain why the population during the DOR monotherapy struggled to acquire mutations before transfer 12.

Many thanks for this comment. We agree with the reviewer that a more detailed population genomic analysis could have provided further insights into the absence of resistance changes in the bacteria from the DOR monotherapy treatment. However, such additional genomics data is not essential for our main conclusions. Firstly, the phenotypic and genomic results are consistent, as both demonstrate a lack of resistance to the DOR treatment. Secondly, the phenotypic and current genomic data only provide an indication of a lower mutation rate towards DOR resistance. Additional population genomic data could confirm it, but not validate it. Such a validation may best be achieved by an independent experimental approach, which directly assesses the rate of DOR resistance mutations. This is the reason why we performed such an independent experimental test, using fluctuation assays. This test indeed confirmed the initial finding. Therefore, we consider our conclusion of a constrained rate of DOR resistance evolution to be valid and justified.

Reviewer #3 (Recommendations for the authors):[…]This study is timely, important, extremely thorough, and perhaps most importantly, it helps to identify a number of new principles to guide the rational design of sequential antibiotic therapies. It represents a monumental effort to thoroughly characterize adaption to 3-drug cycles in several representative examples, an achievement in itself. But as impressively, the authors draw on GLM's to extract meaningful patterns from the web of complex phenotypic data, and in doing so they uncover features of these treatments (e.g. surprising cross-resistance landscapes between drugs from the same class; potential importance of having one drug with low rates of spontaneous resistance) that may guide general design of new therapies. As with all good studies, it raises a number of questions for future work, and I list a few that come to mind below (that authors may wish to respond, perhaps in the discussion, at their discretion). But in general, I have no major concerns with the study at a technical or conceptual level (a rarity!). I believe this is an excellent contribution to the literature that will interest a broad range of readers in evolutionary biology and microbiology.1) To me, one of the most interesting aspects of the study is that the principles identified for designing sequential therapies are not dependent on a unique molecular mechanism, and one could therefore envision multiple drug combinations that achieve similar evolution-slowing effects but do so in very different ways at the molecular level (where resistance has often been studied). Do the authors believe there is some inherent feature of β-lactams-for example, the fact that they compromise the cell wall and promote lysis-that may underlie the success of homogeneous sequential therapies? Or might one expect similar success with, say, triplets of fluoroquinolones or other drugs, as long as the triplet is chosen with similar qualitative features (e.g. low levels of cross-resistance, low spontaneous resistance to at least one drug). Relatedly, are there reasons to believe that certain classes of drugs will be more likely to show those qualitative features? It would be remarkable if, for example, these important features could be predicted based on, say, the known diversity of resistant mutations for the component drugs. In any event, I think one of the striking conclusions from this work is that considerable optimization may be possible without detailed knowledge of molecular mechanism.

We share the enthusiasm of the reviewer that the available drugs offer many currently unsuspected options to optimize antibiotic therapy. We agree that the raised questions would be extremely fascinating to study in more detail. They cannot really be answered based on the current data. They do require future additional studies that systematically assess the ability of bacteria to adapt to a variety of multi-drug sequential therapy as well as the underlying mechanisms. Further work is also required to understand the exact molecular basis of interactions between different β-lactam drugs or the general influence of variation in spontaneous resistance to different antibiotics on the efficacy of treatments with these drugs.

2) The results suggest that the chosen β lactams exhibit negative hysteresis, yet the negative hysteresis does not play a dominant role in slowing resistance (unlike in the authors' previous study with heterogeneous drug combinations). Does this discrepancy arise simply because other effects (e.g. collateral effects) are dominant, so the impacts of negative hysteresis are (by comparison) negligible? Or is there some feature of the negative hysteresis-and perhaps its correlation with the collateral effects-that renders it non-beneficial? I'd venture a guess that in general, there is some underlying relationship between the hysteresis profiles, the collateral profiles, and the timescale of switching that could conceivably be tuned to optimize these therapies. It seems to be an exciting topic for future work.

Many thanks for these interesting comments. Again, the raised questions cannot really be answered in consideration of the current data. There is clearly a need for a more systematic analysis of the relationship between hysteresis, collateral effects, and additionally the rate of antibiotic switches. Our earlier work found a two-component regulator *cpxS* to contribute to negative hysteresis, but only mildly to resistance, possibly suggesting that hysteresis relies on molecular processes that are distinct to those mediating resistance or collateral sensitivity. However, this is only a single case and more examples need to be characterized in detail before general conclusions can be drawn